# A new and inexpensive non-bit-for-bit solution reproducibility test based on time step convergence (TSC1.0)

Hui Wan[1], Kai Zhang[1], Philip J. Rasch[1], Balwinder Singh[1], Xingyuan Chen[1], and Jim Edwards[2]

[1]Pacific Northwest National Laboratory, Richland, WA, USA
[2]National Center for Atmospheric Research, Boulder, CO, USA

*Correspondence to:* Hui Wan (Hui.Wan@pnnl.gov)

**Abstract.** A test procedure is proposed for identifying numerically significant solution changes in evolution equations used in atmospheric models. The test issues a "fail" signal when any code modifications or computing environment changes lead to solution differences that exceed the known time step sensitivity of the reference model. Initial evidence is provided using the Community Atmosphere Model (CAM) version 5.3 that the proposed procedure can be used to distinguish rounding-level solution changes from impacts of compiler optimization or parameter perturbation that are known to cause substantial differences in the simulated climate. The test is not exhaustive since it does not detect issues associated with diagnostic calculations that do not feedback to the model state variables. Nevertheless it provides a practical and objective way to assess the significance of solution changes. The short simulation length implies low computational cost. The independence between ensemble members allows for parallel execution of all simulations thus facilitating fast turnaround. The new method is simple to implement since it does not require any code modifications. We expect that the same methodology can be used for any geophysical model to which the concept of time step convergence is applicable.

## 1 Introduction

The Community Atmosphere Model (CAM, Neale et al., 2010, 2012), like all other general circulation models (GCMs) used for weather and climate prediction and research, is a large body of computer code that solves a system of differential, integral, and algebraic equations. Testing the code to ensure it behaves as expected involves a wide range of efforts that touch upon the formulation of the equations, the solution algorithms, and the software design and implementation. This paper addresses the issue of regression testing, i.e., verifying that results from the model stay the same despite changes in the code or the computing environment. In certain cases, it is possible to achieve this goal by demonstrating that a newly conducted simulation produces bit-for-bit (BFB) identical output compared to a simulation previously certified to be valid. More often, however, software or hardware updates as well as code optimization or refactoring inevitably lead to the loss of BFB reproducibility, in which case a different criterion is needed to declare two simulations as "the same". The large number of equations in an atmospheric GCM and the nonlinearities of the equation set make it a challenging task to define such a criterion.

Since CAM is a climate model, one possibility could be to require that the long-term statistics of the atmospheric motions be representative of the climate simulated by the old code in the old environment (see, e.g., "Condition 3" in Rosinski and

Williamson, 1997). One procedure to make such an assessment could be a "Subjective Independent Examination and Verification by Experts", or SIEVE, that consists of experienced climate modelers performing multi-year simulations and examining many fields of the model output to determine whether the simulated climate has changed or not. This procedure is unsatisfactory due to its subjectivity and the high computational cost, but we speculate this is the most widely used method in many modeling groups. Recently, Baker et al. (2015) developed an Ensemble-based Consistency Test (ECT) as a replacement of SIEVE, which we refer to as CAM-ECT following Baker et al. (2016). CAM-ECT involves first generating a reference ensemble of one-year simulations on a trusted computer with an accepted version and configuration of CAM, and creating a statistical distribution that characterizes the ensemble using principal component analysis (PCA) of the globally averaged annual mean fields. To test a new code or computing environment, a small ensemble of one-year simulations is conducted, and the CAM-ECT tool determines whether the new simulations are statistically distinguishable from the reference ensemble. Compared to SIEVE, CAM-ECT is a major step forward in regression testing since it clearly defines an objective criterion for "pass" or "fail". The use of PCA allows the test diagnostics to include all variables written out by the model, resulting in rather complete code coverage. As demonstrated by Baker et al. (2015) and Milroy et al. (2016), the method is able to detect the impact of parameter changes in the model source code as well as issues in the computing environment. The main limitation of CAM-ECT lies in its computational cost. In the original implementation described by Baker et al. (2015), the reference ensemble consisted of 151 members and the test ensemble included 3 simulations. A follow-up study by Milroy et al. (2016) proposed using 453 simulations from multiple compilers to provide sufficient variability in the reference ensemble. Since the reference ensemble needs to be updated every time a new code version with different climate characteristics is selected as the baseline for further model development (e.g., after a climate-changing bug fix), the large ensemble size can be a substantial burden in computational cost, especially during very active model development phases.

Given that the purpose of the regression testing is to assure the model results stay the same, rather than to provide a descriptive characterization of the simulated physical phenomena, it would be useful to have additional test methods that can give early warnings of unexpected model behavior using computationally inexpensive simulations. The perturbation growth test (hereafter PERGRO) based on the work of Rosinski and Williamson (1997) is an example that assesses the short-term behavior of the model results. PERGRO was originally designed to verify the simulations after a predecessor of CAM was ported to different computers. More generally, the method has been used to verify that code modifications only produced roundoff-level changes in the model results.

The PERGRO test involved comparing one test simulation and two trusted simulations over the course of two model days. Solution differences were quantified by the spatial root-mean-square differences (RMSD) in the temperature field at each time step. The differences between the two trusted simulations were triggered by random temperature perturbations of order $10^{-14}\,\mathrm{K}$ introduced to the initial conditions in one of the simulations. Rosinski and Williamson (1997) established two conditions for the verification of a ported code:

- Condition 1. During the first few time steps, differences between the original and ported code solutions should be within one to two orders of magnitude of machine rounding.

- Condition 2. During the first few days, growth of the difference between the original and ported code solutions should not exceed the growth of the initial perturbation.

It is worth noting that in order for those two conditions to be useful for the intended verification, the model code has to satisfy a "Condition 0":

- Condition 0. During the first few time steps, rounding-level initial perturbations introduced to the original code in the original environment should not trigger solution differences larger than one to two orders of magnitude of machine rounding.

If Condition 0 is violated, it is expected that the ported code will always fail Condition 1 whether there is a porting error or not. In addition, rapid growth of perturbations even in a trusted computing environment can make it difficult to distinguish

differences between trusted solutions from differences between a trusted solution and a problematic test solution, causing misleading fulfillment of condition 2. Therefore, if Condition 0 is violated, Conditions 1 and 2 might no longer be useful for port verification.

When the PERGRO test was originally developed, the physical parameterizations were quite simple, the code was able to satisfy Condition 0, and the test method was robust. As the model became more comprehensive and complex, more rapid growth

of rounding-level initial perturbation was observed. Compromises were made to preserve some utility for the test. For example, in CAM4, the test needed to be performed in an aqua-planet configuration, i.e., without the land surface parameterizations, and with a few (small) pieces of code in the atmospheric physics parameterizations switched off or revised, because those codes were known to be very sensitive to small perturbations. If those pieces of codes were not switched off or revised, perturbations on the trusted machine would grow so rapidly that the RMSD would reach $\mathcal{O}(0.1)$ K over a few timesteps. Disabling the land

interactions and a few pieces of code returned the bulk of the atmospheric model to a configuration where differences between perturbed and unperturbed initial conditions grew substantially more slowly. Most of the time, the RMSD grew at a rate well below one order of magnitude per timestep in a trusted environment. An example is shown in Fig. 1a with the blue curve. With the revised aqua-planet configuration of CAM4, it was still possible to examine solution differences between original and test solutions to see whether they violated Condition 2 for a port verification effort. But with CAM5, initial perturbations grow too

rapidly even in an aqua-planet simulation (Fig. 1a, red curve), making the original PERGRO method no longer useful for port verification.

Rosinski and Williamson (1997) noted that dynamical-core-only simulations typically showed much slower growth of initial perturbation, and this characteristic remains true in newer model versions. For example, using the default configuration of CAM5's spectral element dynamical core (Taylor and Fournier, 2010; Dennis et al., 2012) at 1 ° spatial resolution, the

30 temperature RMSD only reaches $\mathcal{O}(10^{-12})$ K by day 2, suggesting that the rapid growth shown in Fig. 1a is due to the physics parameterizations. Efforts have been made to understand the cause of the rapid growth, and those findings will be detailed in a separate manuscript.[1] Here we provide only a brief description of three causes:

---

[1] Singh B., Rasch, P. J., Wan, H., and Edwards, J.: A verification strategy for atmospheric model codes using initial condition perturbations. To be submitted.

First, the default time step of 1800 s in CAM5 is sizable compared to the characteristic time scales of many physical processes represented by the model, so the increments in the model state during one time step (i.e., the process tendencies times the model time step) are significant, and the differences between a pair of simulations with slightly different initial conditions can also be perceptible. The red and purple curves in Fig. 1b show that when the time step sizes of all model components are changed by a factor of 1800, the solution differences after the same number of time steps also change by a similar ratio. Longer model time steps lead to larger increments from the simulated physical processes, but not necessarily so for software or hardware issues. Therefore the growth of perturbation in a model with long time step can make it difficult to expose solution differences caused by a new computing environment.

The second reason for rapid perturbation growth is related to the fact that the radiation parameterization in CAM5 uses a pseudo random number generator, and the seeds for the generator are chosen from the less significant digits of the pressure field. This effectively introduces state-dependent noise into the numerical solution. The green curve in Fig. 1b shows the differences between a pair of simulations conducted with 1 s time step but with radiation calculated only once at the beginning of the integration. Compared to the purple curve where radiation was calculated every other time step, the solution differences were further reduced by about 3 orders of magnitude. We note that the noisiness from the radiation calculation can be controlled by making the random seeds independent of the model state so that the random series become reproducible from one simulation to another; but more generally, the radiation example also implies that models with state-dependent stochastic parameterizations might feature more rapid perturbation growth than those using deterministic schemes.

The third reason for rapid perturbation growth has to do with particular pieces of code. Two types of examples were discussed in Rosinski and Williamson (1997): (i) an upshift in digit of solution difference resulting from division by a small number, and (ii) if-statements associated with algorithmic discontinuity. We have experienced both types of situations in the CAM5 code, although the specific formulae were different from those given by Rosinski and Williamson (1997). Compared to its predecessors, CAM5 uses modern parameterizations with substantially more detailed description of the atmospheric phenomena, and the model also carries an expanded list of tracers. The increase in model complexity and the corresponding growth in the size of the code substantially increase the chance for similar situations to occur.

The examples shown in Fig. 1b indicate that it is possible to identify reasons for perturbation growth, with the potential to make PERGRO a useful testing method again, although experience shows that such efforts can be rather substantial and time-consuming. We will document that path elsewhere. In the present paper, we describes a strategy that tests a code "as is" so that new parameterizations and code updates can be assessed as soon as they enter the model. The new test procedure is based on the work of Wan et al. (2015) on time step convergence in CAM5. The underlying concept and design considerations are explained in Sect. 2. A first implementation of the test in CAM5 is described in Sect. 3 and evaluated in Sect. 4. Further discussions on the test design and its relationship to other methods are presented in Sect. 5. Conclusions are drawn in Sect. 6.

## 2 Test philosophy

In this section, we start with a clarification of the purpose and scope of the new test method (Sect. 2.1), then proceed to a discussion of the desirable features that guided the design of our test (Sect. 2.2). The underlying concept of the new method is explained in Sect. 2.3.

### 2.1 Purpose and scope

As stated earlier, the topic of this paper is regression testing under circumstances when results from an atmospheric GCM are no longer BFB reproducible. In other words, the testing discussed here aims at substantiating whether results from an atmospheric GCM stay the same after supposedly minor code modifications or computing environment changes. By "minor code modifications" we mean code refactoring, optimization of the computational efficiency, or any other code changes that might alter the sequence of computation but still solve the same set of equations using the same mathematical algorithms. Computing environment changes refer to any changes in the hardware or software configuration in which the model code is compiled and executed. Two factors need to be considered when designing a method for regression testing: (i) the variables that represent the outcome of a simulation, and (ii) a criterion for declaring two simulations as "the same". In the present paper, we consider the outcome of a simulation unchanged if the numerical solution is found to have the same time stepping error relative to a reference solution obtained with a previously verified code and computing environment. The details are explained later in Sect. 2.3. The reasoning behind our choice for element (ii) is explained below.

From the perspective that a GCM is a suite of algorithms solving a large set of differential, integral, and algebraic equations, the physical quantities (model variables) calculated by the code can be sorted into 3 categories:

I. Prognostic and diagnostic variables whose equations are coupled to one another such that any change in variable $A$ will, within one time step or after multiple time steps, affect variable $B$ in this same category. Examples in this category include basic model state variables like temperature, winds, and humidity, as well as quantities calculated as intermediate products in a parameterization, for example the aerosol water content (which affects radiation and eventually temperature), and the convective available potential energy (which affects the strength of convection hence temperature and humidity).

II. Prognostic variables that are influenced by type-I variables but do not feedback to them. An example could be passive tracers carried by the model to investigate atmospheric transport characteristics (e.g., Zhang et al., 2008; Kristiansen et al., 2016)

III. Diagnostic quantities calculated to facilitate evaluation of a simulation, but do not feedback to type I or type II. Examples include the daily maximum 2-m temperature, the total ice-to-liquid conversion rate in the cloud microphysics parameterization (which is calculated merely for output in CAM5), and any variable specific to the COSP simulator package (Bodas-Salcedo et al., 2011).

We take the standpoint that the essential characteristics of the simulated atmospheric phenomena are determined and represented by type-I variables. If instantaneous and grid-point values are monitored, any significant solution change should be detectable through the monitoring of a single variable in type I, per definition of that variable type, as long as the simulations are long enough for the impact to propagate and evolve to a discernable signal in that monitored variable. On the other hand, since we are taking a deterministic perspective here, the simulations need to be sufficiently short to avoid chaos.

Based on the reasoning above, the test diagnostics of our new method are calculated from a small set of prognostic variables of type I. The use of multiple variables is meant to help increase the sensitivity of the test (decrease the chance of failing to detect a significant solution change), since bugs or issues associated with a specific piece of code might take longer time to cause discernable solution differences in one variable than in another. In Sects. 3 and 4 where we describe and evaluate the first implementation of our method in CAM5, the monitored variables include a few basic atmospheric state variables plus aerosol and hydrometeor concentrations. We note that this choice of variables can be further evaluated or tailored to meet the user's needs. The test method can also be extended to include variables of type II, but cannot be used on type-III variables or diagnostic variables in type I, because the concept of time step convergence does not apply. This means our test does not provide a full coverage of all code pieces in the model. For example, bugs in the implementation of a satellite simulator or other "diagnostic-only" calculations would not be detected by our test. Issues in software functionalities that are not exercised during the simulations, e.g., the reading and writing of restart files, would not be caught, either. We acknowledge that the proposed test method is not exhaustive; but given its simplicity, low computational cost, and the effectiveness demonstrated in Sect. 4, we believe it is a practical and promising method for assessing the magnitude of solution differences in complex models.

## 2.2 Desirable features

Given the continuously growing complexity of the modern atmospheric GCMs and the need by large groups of model developers and users to perform regression testing routinely (e.g. on a daily basis), it is desirable to have test procedures that have the following features:

1. Objective;

2. Easy to perform and automate;

3. Requiring no or minimum code modifications;

4. Exercising the entire model in its "operational" configuration;

5. Also applicable to a subset of the code thus useful for debugging;

6. Capable of detecting changes in both global and/or regional features of the simulations;

7. Insensitive to roundoff differences associated with changes in the order of accumulations or associative operations, etc;

8. Computationally efficient.

The CAM-ECT of Baker et al. (2015) fulfills criteria 1–4 and 7, and partly 5. For criterion 5, we expect CAM-ECT to be capable of isolating issues associated with variables of type II or III (cf. Sect. 2.1) through systematic elimination of model output variables from the test diagnostics (Milroy et al., 2016). Bugs associated with type-I variables would be more difficult to pinpoint: since all variables in this type are inherently coupled, we expect that any substantial change in one equation would have affected all the type-I variables after a year of model integration. One-year simulations might also be challenging for a code that is still in debugging stage thus numerically unstable for long simulations. The use of global annual averages by CAM-ECT might lead to difficulty in detecting changes in small-scale features (criterion 6). For example, Baker et al. (2015) noted that CAM-ECT did not identify the impact of a perturbed horizontal diffusion parameter as "climate-changing" (see case NU discussed in Sect. 4.3 therein). On the other hand, since a large number (120) of model output variables are used in CAM-ECT and the simulations are relatively long thus allowing ample time for the impact of a bug or system issue to evolve and propagate, the chance of missing a climate-changing feature (i.e. getting a false "pass") is relatively small. The main limitation of CAM-ECT lies in its computational cost (criterion 8), as already mentioned in Sect. 1.

The PERGRO test of Rosinski and Williamson (1997) fulfills criterion 7 per design. The use of 2-day simulations translates to very low computational cost thus fulfilling criterion 8. the method also satisfies criteria 2, 3, 5, and 6. The aqua-planet setup with a few test-specific code changes leads to a configuration that is very close to the full version of the atmosphere model (criterion 4). The interpretation of the perturbation growth test has some subjectivity (criterion 1), since there is not a quantitative criterion regarding how close the new RMSD curve should resemble the reference curve. However, the developers' experience was that when a simulation fails the test, "it generally fails spectacularly, i.e., the difference curve will exceed the perturbation curve by many orders of magnitude within a few model timesteps" (http://www.cesm.ucar.edu/models/cesm1.0/cam/docs/port/pergro-test.html). Therefore objectivity is also not a major weakness of the PERGRO test. The main – and also critical – difficulty with the method is that it is ill-suited for CAM5 because the "Condition 0" needed by the test strategy has now been violated.

The new test proposed in this paper aims at satisfying all the 8 features listed above. It keeps the deterministic spirit of PERGRO to achieve an early detection of solution differences thus saving computational time. Ensemble simulations are conducted to take into account the internal variability of the atmospheric motions. The test design was inspired by the results of Wan et al. (2015), as explained below. In the remainder of the paper, we will refer to the new test method as the Time Step Convergence (TSC) test.

## 2.3 Time step convergence (TSC)

Wan et al. (2015) evaluated the short-term time step convergence in CAM5 for the purpose of quantifying and attributing numerical artifacts caused by time integration. Starting from the same initial conditions, a series of $1\,\mathrm{h}$ simulations were conducted using time step sizes ranging from $1\,\mathrm{s}$ to $1800\,\mathrm{s}$. The numerical solution with $\Delta t = 1\,\mathrm{s}$ was viewed as the proxy "truth", and the time stepping error associated with a longer step size was defined as the RMSD between instantaneous 3D temperature fields after $1\,\mathrm{h}$ of model integration. To take into account possible flow-dependencies of the numerical error, the exercise was repeated using initial conditions sampled from different months of a previously conducted multi-year simulation,

following the idea of Wan et al. (2014). A linear regression was then applied between the ensemble mean $\log_{10}(\text{RMSD})$ and $\log_{10}(\Delta t)$. The regression coefficient gives the time step convergence rate. Experience so far indicates that the diagnosed convergence rate is rather insensitive to the choice of initial conditions (cf. Sect. 3.2 for further discussion).

In Fig. 2, the 12-member ensemble mean temperature RMSD in the default CAM5.3 model ("CTRL") is shown with blue circles, and the $\pm\sigma$ ranges are shown by vertical bars. Here $\sigma$ denotes the ensemble standard deviation. The blue regression line indicates a convergence rate close to 0.4. It is important to emphasize that this regression line corresponds to the *self-*convergence, i.e., the convergence towards a solution produced with the same code and a very small step size. When the code is not exercised correctly, or when the model equations have changed because of parameterization update or parameter tuning, convergence towards the original reference solution should no longer be expected. This is the key hypothesis on which our new test method is based.

To demonstrate this point, Fig. 2 also shows results from simulations conducted with a modified parameter in the physics package. Specifically, the grid-box mean relative humidity threshold for the formation of high-level clouds, a parameter called cldfrc_rhminh in the large-scale condensation scheme of Park et al. (2014), was changed from 0.8 to 0.9. This parameter change was used in Baker et al. (2015) in the evaluation of CAM-ECT, and we label it "RH-MIN-HIGH" following that study. The RMSD calculated against a new reference solution using cldfrc_rhminh = 0.9 and $\Delta t = 1\,\text{s}$ is shown in green in Fig. 2. The self-convergence of the modified model turns out to be very similar to the self-convergence in the original model. This is expected, and also consistent with the concept of self-convergence since no structural changes (e.g. parameterization or numerical algorithm modifications) have been introduced into the model. However, when the RMSD of the RH-MIN-HIGH simulations are calculated against the $1\,\text{s}$ simulations of CTRL, the RMSD values appear to be considerably larger at smaller step sizes. The discrepancies – caused by the parameter change – far exceed the ensemble spread of the reference solutions. The divergence of the red and blue convergence pathways in Fig. 2 provides a proof of concept that the model's time step convergence behavior can be used as a metric to detect significant changes in the numerical solution. In Fig. 2, the RMSD is shown for a range of step sizes for a better illustration of the concept. In practice, anomalous RMSD at one step size will be sufficient to flag a code or computing environment as failing the expectation that they provide the same numerical solution as the reference code or environment does, although the identification of a "true anomaly" requires an ensemble of independent simulations, which we will demonstrate in Sect. 3.2.

Fig. 2 also indicates that the RMSDs calculated both ways are hardly distinguishable at the default step size, suggesting that the impact of the parameter change is smaller than or similar to the time integration error, at least for this prognostic variable and at the chosen time scale (1 h). If we had introduced larger changes in the model, e.g., by changing cldfrc_rhminh to 0.999 instead of 0.9 from the default value of 0.8, or by replacing a certain parameterization by a different scheme, the impact might be more visible at the default step size. In contrast, if the parameter change were smaller, e.g., from 0.8 to 0.82 instead of 0.9, the red and blue convergence pathways in Fig. 2 might not diverge until a step size on the order of a few seconds. In order to establish a highly sensitive regression test that can detect very small solution changes, it would be desirable to find a time step size that corresponds to very small numerical error. The shortest possible step size for CAM5.3 simulations is $1\,\text{s}$; this is the shortest possible interval at which the dynamical core and the various parameterized physical processes interact with each

other, and also the shortest step size the coupler can handle for the coupling between different model components (atmosphere, land, ocean, sea ice, etc.). Hence the new TSC test uses the RMSD between a pair of simulations with 2 s and 1 s time steps as the metric for assessing the magnitude of solution changes.

In the study of Wan et al. (2015), simulations with shortened time step sizes were conducted with all physics parameterizations calculated every time step except for radiation which was called only once (i.e., with a 1 h step size, cf. Table 1 in Wan et al., 2015). The simulations shown in Fig. 2 followed the same design, but we also repeated the simulations with radiation calculated every other time step (as in the default model). The results were hardly distinguishable from Fig. 2 (not shown), suggesting that the calling frequency of radiation does not change the convergence property of the CAM5 model. When describing the TSC implementation in the next section, we propose to calculate radiation every other time step so that the time step ratio is kept the same among all model components. In Sect. 5 we also present results from simulations with radiation calculated only at the first time step, and discuss the impact of noisy parameterization on the TSC results.

We also note that in the earlier study of Wan et al. (2015), convergence analysis was done not only with the full CAM5 model, but also using configurations that exercised the dynamical core plus one parameterization or parameterizations group at a time, e.g., deep convection, shallow convection, large-scale condensation, or the stratiform cloud microphysics, as an attempt to find out which of those parameterizations led to the convergence rate of 0.4 instead of 1 in the full model. Additional simulations were conducted using the dynamical core plus a simple saturation adjustment scheme or with the cloud microphysics parameterization of CAM5 but with the formation and sedimentation of rain and snow turned off (cf. Fig. 3 in Wan et al., 2015). Those simulations revealed different convergence rates and time step sensitivities associated with different components of the model code. We expect that this strategy of breaking down the code into small exercisable units could be used to pinpoint bugs when, e.g., a code refactoring effort leads to solution differences that are unexpectedly large according to the TSC test. In other words, we expect the TSC method to fulfill feature 5 listed in Sect. 2.2. Future work is planned to evaluate TSC's utility for that purpose.

## 3 Implementation

In this section we first give a brief overview of the CAM5 model (Sect. 3.1), emphasizing only on the aspects that are directly relevant for the technical implementation of the TSC test. The test procedure is then described in detail in Sect. 3.2

### 3.1 CAM5.3 overview

The global climate model used in this paper is CAM5.3 (Neale et al., 2012) with the spectral element dynamical core (Taylor and Fournier, 2010; Dennis et al., 2012). The dynamical core solves a hydrostatic version of the fluid dynamics equation, with surface pressure (PS), temperature (T), and horizontal winds (U, V) being the prognostic variables. In addition, the model includes budget equations for specific humidity (Q), as well as the mass and number concentrations of the stratiform cloud droplets (CLDLIQ, NUMLIQ) and ice crystals (CLDICE, NUMICE). The time evolution and spatial distribution of water vapor and hydrometeors are affected by resolved-scale transport and by subgrid-scale moist processes such as turbulence,

convection, and cloud microphysics. Those subgrid-scale processes provide feedback to the thermodynamical state of the atmosphere through latent heat release. CAM5.3 also has a Modal Aerosol Module (MAM, Liu et al., 2012; Ghan et al., 2012) that represents the life cycle of 6 aerosol species: sulfate, black carbon, primary organic aerosols, secondary organic aerosols, sea salt, and mineral dust. The size distribution of the aerosol population is mathematically approximated by a few log-normal modes. In this study we used the 3-mode version of MAM, thus the model's prognostic variable set also includes the particle number concentrations of the 3 modes (num_a1, num_a2, and num_a3, for the accumulation mode, Aitken mode, and coarse mode, respectively), and the mass concentrations of each aerosol species in each mode.

In the present paper we use the FC5 component set of the model, meaning that the model is configured to run with interactive atmosphere and land, prescribed climatological sea surface temperature and sea ice cover, and with the anthropogenic aerosol and precursor emissions specified using values representative of the year 2000.

## 3.2 Test procedure

The basic idea of the TSC test is to perform control and test simulations with a $2\,\mathrm{s}$ time step, calculate their RMSDs with respect to reference simulations conducted with the control model with a $1\,\mathrm{s}$ time step, then determine whether the RMSDs of the control and test simulations are substantially different.

For a generic prognostic variable $\psi$, we define

$$\mathrm{RMSD}(\psi) = \left\{ \frac{\sum_i \sum_k w_i \left[ \Delta\psi(i,k) \right]^2 \Delta\bar{p}(i,k)}{\sum_i \sum_k w_i \, \Delta\bar{p}(i,k)} \right\}^{1/2}, \tag{1}$$

$$\Delta\psi(i,k) = \psi(i,k) - \psi_r(i,k), \tag{2}$$

$$\Delta\bar{p}(i,k) = \left[ \Delta p(i,k) + \Delta p_r(i,k) \right]/2. \tag{3}$$

Here $\Delta p(i,k)$ denotes the pressure layer thickness at vertical level $k$ and cell $i$, and $w_i$ is the area of cell $i$. Subscript $r$ indicates the reference solution. This formulation of RMSD follows the work of Rosinski and Williamson (1997).

Time step size affects the numerical solution at every time step and every grid point, while certain atmospheric processes might occur in isolated regions thus impacting only a limited number of grid points during very short simulations. Consequently, subtle but systematic solution changes can be masked by the model's time stepping error and can be difficult to detect. To help address this challenge, we calculate RMSDs for $N_{\mathrm{dom}} = 2$ domains, i.e., land and ocean, separately. This is a practical and somewhat arbitrary choice aiming at increasing the sensitivity of the TSC test.

As for the physical quantities, the results shown in the present paper include RMSD of $N_{\mathrm{var}} = 10$ prognostic variables: V, T, Q, CLDLIQ, CLDICE, NUMLIQ, NUMICE, num_a1, num_a2, and num_a3 (i.e. the meridional wind field, temperature, specific humidity, grid-box mean mass and number concentrations of the stratiform cloud droplets and ice crystals, and the particle number concentrations of the three log-normal modes that describe the aerosol size distribution, respectively). This selection of prognostic variables is motivated by an emphasis on atmospheric circulation, thermodynamics, clouds, and aerosols. The mass concentrations of aerosol species are not included, because it is unlikely that a perturbation will change the aerosol mass concentrations without affecting the number concentrations after multiple steps of integration. Additional variables of type I

defined in Sect. 2.1 can be added to the list, and a longer variable list might help increase the sensitivity of the test. Type-II variables can also be added if the user wishes to cover the respective code pieces. The TSC method is flexible in this regard, although we emphasize again that only prognostic variables of type I and type II can be included in the list. The concept of time step convergence does not apply to variables that are not calculated using an evolution equation.

The test procedure includes three steps. Steps 1 and 2 are needed every time a new baseline model with different solution characteristics is established. Between such baseline releases, only step 3 is needed for the testing of a new code version or computing environment.

    **Step 1:** Create an $M$-member simulation ensemble with a control version of the model in a trusted computing environment, using $1\,\text{s}$ time step for a simulation length of $X$ minutes. These are considered the *reference solutions*. The independent
members are initialized on January 1, 00Z using model states sampled from different months of a previously performed climate simulation, with non-zero concentrations for water vapor, hydrometeors, aerosols, and all other tracers that the model carries. Save the 3D instantaneous values of the $N_{\text{var}}$ prognostic variables listed above, plus the values of surface pressure and land fraction, all in double precision, after a model time of $t$.

    **Step 2:** Obtain an $M$-member ensemble using the same initial conditions as in step 1, again with the control model in a
15 trusted computing environment, but using a 2 s time step. Compute the RMSD using Eq. (1) for each pair of simulations that started from the same initial conditions. The resulting RMSDs at time $t$ are denoted as $\text{RMSD}_{\text{trusted},t}$.

    **Step 3:** Repeat Step 2 with a modified code or in a different computing environment. Compute the RMSDs with respect to the reference solutions created in Step 1, and denote the results at model time $t$ as $\text{RMSD}_{\text{test},t}$. Now define

$$\Delta\text{RMSD}_{t,j,m} = \text{RMSD}_{\text{test},t,j,m} - \text{RMSD}_{\text{trusted},t,j,m} \quad (m = 1, \cdots, M; \; j = 1, \cdots, N_{\text{var}} \times N_{\text{dom}}), \tag{4}$$

and denote the $M$-member ensemble mean by $\overline{\Delta\text{RMSD}}_{t,j}$. For each prognostic variable and domain (i.e. each $j$), we assume the ensemble mean of $\Delta\text{RMSD}_{t,j,m}$ is a random variable $\mu_{t,j}$. The students $t$-test is performed on the null hypothesis that $\mu_{j,t}$ is statistically zero, with the alternative hypothesis of $\mu_{j,t} > 0$. A one-sided test is used here because of the concept of self-convergence explained in Sect. 2.3: When bugs are introduced, or when the code is compiled or executed incorrectly, the simulation will not solve the originally intended equations, thus will not converge to the reference solutions produced by the
original code or environment. Let us use the symbol $S_{\text{ori},1s}$ to denote the reference solution of the original equation set obtained with a 1 s time step, and use $S_{\text{test},2s}$ to denote a test simulation conducted with the new equation set using a 2 s time step. The RMSD calculated in TSC is the root-mean-square of $(S_{\text{test},2s} - S_{\text{ori},1s})$ which can also be expressed as

$$(S_{\text{test},2s} - S_{\text{test},1s}) + (S_{\text{test},1s} - S_{\text{ori},1s}) \tag{5}$$

The difference in the first pair of parentheses in (5) measures the time-step sensitivity of the solution of the new equation set,
while the difference in the second pair of parentheses measures the discrepancy between the reference solutions of the old and new equation sets. By using a one-sided test, we assume that the second difference will be non-negligible, and that the two differences will not incidentally compensate each other to result in values of $(S_{\text{test},2s} - S_{\text{ori},1s})$ that are systematically smaller than $(S_{\text{ori},2s} - S_{\text{ori},1s})$. The validity of this assumption can be evaluated in the future by comparing TSC results using one-sided and two-sided tests.

In the present paper we use a one-sided $t$-test. The $j$th variable at time $t$ fails the TSC test if the null hypothesis is rejected, i.e., if

$$\mathcal{P}\left(\mu_{t,j} > \overline{\Delta \text{RMSD}}_{t,j}\right) < \mathcal{P}_0 , \tag{6}$$

where $\mathcal{P}$ stands for probability and $\mathcal{P}_0$ is an empirically chosen threshold. If Eq. (6) turns out to be true for any $j$, or in other words,

$$\mathcal{P}_{\text{min},t} = \min_{j=1,N_{\text{var}} \times N_{\text{dom}}} \left[ \mathcal{P}\left(\mu_{t,j} > \overline{\Delta \text{RMSD}}_{t,j}\right) \right] < \mathcal{P}_0 , \tag{7}$$

then the ensemble fails the TSC test at time $t$.

In case the test and control simulations only contain insignificant differences, $\mathcal{P}_{\text{min},t}$ is expected to be relatively large during the $X$ minutes of integration, but can still get small values by chance, thus appearing like a random variable. In case a bug or software/hardware issue causes substantial solution differences, it is expected that $\mathcal{P}_{\text{min},t}$ will show very small values after a certain time of spin-up. We use this distinction to determine an overall pass or fail for a test ensemble. In order to fully automate the test procedure, a quantitative criterion is needed to describe this distinction. For simplicity and as a preliminary choice, we propose to fail a test ensemble if $\mathcal{P}_{\text{min},t} < \mathcal{P}_0$ for all output steps in a time window $[X_0, X]$, where $X$ is the total simulation length and $X_0$ is the spin-up time. The use of multiple time steps in the overall pass/fail criterion reflects our perspective of viewing the model integration as a time evolution problem. We note that the typical values of $\mathcal{P}_{\text{min},t}$ depend on the number of monitored variables (i.e., larger $N_{\text{var}} \times N_{\text{dom}}$ can result in smaller $\mathcal{P}_{\text{min},t}$ in a statistical sense), hence $\mathcal{P}_0$ needs to be determined empirically for a given $N_{\text{var}} \times N_{\text{dom}}$. Ideally $\mathcal{P}_0$ should be small enough to reduce the chance of false positive (i.e., insignificant solution differences being assigned a "fail"), and large enough to reduce the chance of false negative (i.e., subtle but systematic solution differences being assigned a "pass"). In the present paper we have made an empirical and somewhat arbitrary choice of

$$\mathcal{P}_0 = 0.5\%, \quad X_0 = 5 \,\text{min}, \quad X = 10 \,\text{min} . \tag{8}$$

Further evaluation of this choice and possible improvement of the overall pass/fail criterion are topics of future work. In the next section, we present results from $30 \,\text{min}$ simulations with the test diagnostics calculated every minute to reveal the time evolution of $\mathcal{P}_{\text{min},t}$.

$M = 12$ ensemble members are used in this study. One set of initial conditions is sampled from each month of the year to obtain a reasonable coverage of the seasonal variations in the atmospheric circulation, clouds, and aerosol life cycle. The purpose is to account for possible flow-dependencies of the numerical error. The need for an ensemble is demonstrated in Fig. 3 where the normalized $\Delta \text{RMSD}$ of selected variables is shown for individual ensemble members after $5 \,\text{min}$ of integration in an experiment with a modified parameter in the deep convection parameterization over land ("CONV-LND", following Baker et al., 2015; cf. Table 1 and Sect. 4.2 for further details). Passing and failing variables are indicated by dashed and solid lines, respectively. Ocean and land are shown in separate panels using different scales for the y-axes. The values of $\Delta \text{RMSD}_{t,j,m}$ have been normalized by the mean RMSD of the trusted ensemble, i.e., by $\overline{\text{RMSD}}_{\text{trusted},t,j}$. Our exploration has indicated that,

due to the complexity and nonlinearity of the model equations, the values of $\Delta$RMSD of a passing variable from individual ensemble members often are distributed around zero (Fig. 3a). Therefore a single positive $\Delta\text{RMSD}_{t,j,m}$ cannot be viewed as sufficient evidence of non-convergence towards the reference solution. The magnitude of a positive $\Delta\text{RMSD}_{t,j,m}$ is not a good indicator, either, as Fig. 3b shows that even after normalization, a failing variable (e.g. NUMICE in Fig. 3b) can still have small albeit consistently positive $\Delta$RMSD, while a passing variable (e.g. Q in Fig. 3b) may occasionally show large deviations from zero. We have not yet explored the dependence of the test results on the ensemble size, but plan to do so in the future. Furthermore, while we currently apply a $t$-test to determine whether the ensemble *mean* $\Delta$RMSD is equal to or larger than zero, more advanced methods might help to better characterize the ensemble *distribution*.

For all the simulations presented in this paper, the initial conditions were sampled from the first year (after 6 months of spin-up) of a previously conducted 5-year simulation. The decision of using the first year was arbitrary. In our experience, climate simulations of 1–5 years are frequently carried out during model development or evaluation, making such initial conditions easy to obtain. The two features we had in mind when choosing the initial conditions were that (i) they contain reasonably spun-up values for the model state variables (e.g., not all zeros or spatially constant values for the hydrometeors or aerosol concentrations), and (ii) they represent synoptic weather patterns in different seasons. The initial conditions do *not* need to represent well-balanced states in the quasi-equilibrium phase of a multi-year climate simulation. In fact, the default model time step of 1800 s was used when creating the initial conditions for this study, while the control and test simulations in TSC used a 1 s or 2 s time step, so the model state was certainly not well-balanced during those TSC simulations. Also notice that while model states from different seasons were used for initialization, all ensemble members started on January 1, 00Z for simplicity of the simulation and postprocessing workflow, which also led to initial imbalances. Such imbalances are considered harmless since the purpose of the numerical integration is regression testing rather than faithfully simulating the atmospheric motions in the real world. We expect that the same set of initial conditions can be used after answer-changing code baselines are established – until a point when the list of prognostic variables in the model becomes substantially different. Then it would be useful to regenerate the initial conditions, and rethink which variables should be included in the test diagnostics.

## 4 Numerical experiments

Numerical simulations were carried out under a number of scenarios (test cases) to help characterize $\mathcal{P}_{\min,t}$ and evaluate the TSC method. A reference ensemble with a 1 s time step and a trusted ensemble with a 2 s time step were obtained on the supercomputer Titan at the Oak Ridge Leadership Computing Facility using the Intel compiler version 15.0.2 with optimization level -O2. Various test simulations were then conducted in three groups (Table 1).

Group ENV used the same code as in the reference ensemble but with different computers, compilers, or optimization levels:

- PGI compiler version 15.3.0 with -O2 on Titan ("Titan-PGI");

- Intel compiler version 15.0.0 with -O2 on Yellowstone (ark:/85065/d7wd3xhc) at the Computational and Information Systems Laboratory of the National Center for Atmospheric Research ("YS-Intel15-O2");

– Intel compiler version 15.0.0 with -O3 on Yellowstone ("YS-Intel15-O3").

Titan-PGI and YS-Intel15-O2 are supported environments for CAM5.3, in which the simulations are expected to pass the TSC test. The YS-Intel15-O3 case has been found by Baker et al. (2015) to produce incorrect answers, and is expected to fail TSC. (We note that such incorrect answers are produced only when the model is compiled without the "-fp-model" flag. In contrast, if the "-fp-model source" flag is applied to the Fortran code, and the "-fp-model precise" is applied to the C code, the -O2 and -O3 optimization options will produce BFB identical results when CAM5.3 is compiled on Yellowstone with Intel 15.0.0.) We do not include here results from computers that produced borderline pass/fail results in CAM-ECT (e.g., Mira at the Argonne National Laboratory and Bluewaters at the University of Illinois). Valuable investigations have been made by Milroy et al. (2016), but those cases still need further investigation and characterization.

Group MOD consists of two code modification cases from Milroy et al. (2016) that were motivated by optimization of the computational performance:

– In the Division-to-multiplication ("DM") case, division by a time-invariant array was replace by multiplication of the inverse at one place in the dynamical core (cf. Sect. 3.2 in Milroy et al., 2016). This case has been found by CAM-ECT to produce a model climate that is statistically consistent with the reference ensemble. We expect the TSC test to produce a "pass" result;

– In the Precision ("P") case, a subroutine in the physics suite for calculating the saturation vapor pressure over water using the Goff-Gratch formula was changed from double-precision to single-precision. This modification has also been found by CAM-ECT to produce consistent climate, but we put "unknown" in Table 1 for the expected outcome of TSC due to the deterministic nature of the TSC method and the use of double-precision output in the calculation of the test diagnostics.

In group PAR, we repeated all the parameter perturbation experiments presented by Baker et al. (2015), where one parameter in CAM5's physics package was modified in each experiment. All but one case failed CAM-ECT, the exception being the NU case in which the numerical diffusion in the dynamical core was changed by about $10\%$. Baker et al. (2015) pointed out that CAM-ECT gave an unexpected but understandable "pass" flag in this case, because CAM-ECT monitored the global mean values that were not directly affected by the numerical horizontal diffusion. We expect the TSC test to assign "fail" to all cases in this group, including NU, since TSC compares the instantaneous grid-point values of the prognostic variables, thus is expected to be capable of detecting solution changes at all spatial scales resolved by the model. All simulations in groups MOD and PAR were conducted on Titan using the default Intel compiler version and optimization level (15.0.2-O2).

### 4.1 Evolution of $\mathcal{P}_{\min,t}$

To understand the initial evolution of $\mathcal{P}_{\min,t}$, we conducted 30 min simulations and calculated the test diagnostics after every minute. Fig. 4 shows the time series of $\mathcal{P}_{\min,t}$ using a linear scale in panel (a) and a logarithmic scale in panel (b). Two distinct types of behavior can be seen in the figure. In test scenarios where solution differences were thought to be insignificant,

$\mathcal{P}_{\min,t}$ resembles random perturbations around mean values of a few percent. The value at a particular time instance can fall below $1\,\%$, but returns to larger values at later time steps (Fig. 4a). In all test scenarios with modified model parameters, the values of $\mathcal{P}_{\min,t}$ are distinctly closer to zero (Fig. 4a). The time series either show a clear decrease in the first $10\,\text{min}$ and considerably slower changes afterwards (e.g., CONV-LND and NU in Fig. 4b), or start with very low probabilities already and show relatively small changes during the integration (e.g., DUST and FACTIC in Fig. 4b).

The dashed gray lines in Fig. 4 indicate the threshold we chose for assigning an overall "pass" or "fail" to a test ensemble (Eq. 8). The test scenarios that were expected to produce insignificant (significant) solution differences indeed pass (fail) the TSC test. The Precision ("P") case of unknown outcome also passes the TSC test, giving a result consistent with that from CAM-ECT. The two rightmost columns of Table 1 show the values of $\mathcal{P}_{\min,t}$ at $t = 5\,\text{min}$ or averaged between $5\,\text{min}$ and $10\,\text{min}$. Both the instantaneous and averged probabilities are orders of magnitude smaller in the failing cases than in the passing cases.

## 4.2 Results at $5\,\text{min}$

We now take a closer look at the test diagnostics at a single time instance. In Fig. 5, the statistical distributions of $\mu_{t,j}$ (the mean $\Delta$RMSD) estimated from the 12-member ensembles are shown at $t = 5\,\text{min}$ for the individual prognostic variables and domains for four test cases. The values are normalized using the corresponding mean RMSD of the trusted ensemble, i.e., $\overline{\text{RMSD}}_{\text{trusted},t,j}$. The dots indicate the observed ensemble mean (i.e. $\overline{\Delta\text{RMSD}}_{t,j}$), and the filled boxes indicate the $\pm 2\sigma$ range of the mean. The left end of an unfilled box shows the threshold value corresponding to $\mathcal{P}_0 = 0.5\,\%$ in the one-sided $t$-test. Red and blue indicate fail and pass, respectively, according to the criterion defined by Eq. (6). Notice that the x-axes in the subpanels of Fig. 5 are shown in different scales. The normalized mean RMSD differences between the P ensemble and the trusted ensemble are small, on the order of 0.1 or smaller, and the value of 0 lies within the $\pm 2\sigma$ range of the observed $\overline{\Delta\text{RMSD}}_{t,j}$ for all the $N_{\text{var}} \times N_{\text{dom}}$ variables (Fig. 5a). In contrast, the YS-Intel15-O3 case (which is known to produce incorrect solutions) is associated with typical RMSD differences around 1. The large number of failing variables (16 out of 20) and the very small $\mathcal{P}_{\min,t}$ ($1 \times 10^{-11}\,\%$) indicate a clearly failing case.

The test case with a modified dust emission factor (DUST) was expected to be challenging for the TSC method. In any model day, the emission only occurs at a very small fraction of the dust source areas. Dust particles emitted from the surface can only be transported over a short distance during the few-minute simulation time, and the impact on meteorological conditions through the absorption and/or scattering of radiation is also limited. Hence it is unlikely that the solution differences can be seen in the global temperature RMSD. This was the reason that motivated us to use multiple prognostic variables and to separate land and ocean when defining the test diagnostics. The results shown in Fig. 5c confirm our expectation, as only 1 out of the twenty $\overline{\Delta\text{RMSD}}_{j,t}$ is significantly larger than zero. The DUST experiment should nevertheless be considered a clearly failing case since the failing variable (num_a3 over land) is indeed the physical quantity that is most directly affected by dust emission, and the large $\overline{\Delta\text{RMSD}}_{j,t}$ corresponds to a very small $\mathcal{P}\left(\mu_{j,t} > \overline{\Delta\text{RMSD}}_{j,t}\right)$ of $0.0019\,\%$ (cf. Table 1).

The CONV-LND case is challenging for similar reasons. Here the coefficient that controls the conversion of cloud condensate to precipitation was modified for deep convection over land. With a smaller value for zmconv_c0_lnd, we expect to have more

cloud condensate detrained by deep convection, which can lead to changes in the mass and number concentrations of ice crystals in stratiform clouds. Failing results are indeed seen in these two variables (Fig. 5d). The anomalous result in num_a2 is likely related to the removal of aerosol particles by convective precipitation. Since deep convection over land happens in limited areas, and the natural variability is very strong, it is not surprising that $\overline{\Delta \mathrm{RMSD}}_{j,t}$ of the other variables are not yet significantly larger than zero after 5 min of integration.

As mentioned earlier, CAM-ECT assigned a "pass" to the NU case but we expect the TSC result to be a "fail". The respective time series in Fig. 4b reveals $\mathcal{P}_{\mathrm{min},t}$ values below $10^{-4}$ % after 3 min of integration. At 5 min, there are a total of 6 variables with $\mathcal{P}_{t,j} < 0.5\,\%$; the four variables with lowest probabilities are ocean-mean meridional wind, land-mean meridional wind, ocean-mean temperature, and ocean-mean specific humidity. The small minimum probability and the combination of the failing variables provide confidence in the "fail" result of the NU case.

### 4.3 Computational cost

Based on the results shown above, we propose a version 1.0 implementation of the TSC test that uses 12-member 10 min simulations. As such, the computational cost of obtaining an ensemble of reference solutions (using 1 s time step) plus an ensemble of trusted solutions (using 2 s time step) is similar to conducting a single 7.5-month simulation using the default model time step (30 min). For the testing of a new code or computing environment, the cost of conducting 12 simulations using a 2 s time step is similar to that of a 75-day simulation performed using the default time step. Compared to the CAM-ECT which uses 151 to 453 one-year simulations in the reference ensemble and 3 one-year simulations in the test ensemble, the TSC test is a factor of several hundred cheaper to obtain the reference simulations, and a factor of 15 cheaper to test a new code or environment.

The TSC method also allows for very fast test turnaround since the ensemble simulations can be conducted in parallel. On Titan we used 512 MPI processes for each simulation and often submitted 12 simulations to the Portable Batch System (PBS) in three 128-node batch jobs. The wall clock time for finishing a single 10 min simulation with 2 s time step was about 10 min; the entire set of 12 simulations was often completed in 30 min after submission. The time between first job submission and last job completion rarely exceeded a few hours.

### 5 Discussion

In this paper we have presented evidence to demonstrate that the concept of time step convergence can be used to assess the magnitude of solution difference in the CAM model. Future work will be useful to explore the following topics:

### 5.1 Test setup

The TSC test procedure described in this paper has multiple parameters that can be modified: (1) ensemble size, (2) initialization strategy (e.g., simulation start time), (3) time step sizes, (4) integration length, (5) prognostic variables and model sub-domains

included in the calculation of test diagnostics, and (6) the pass/fail criterion. Results presented in the previous section indicate that given (1)-(3), the choices for (4)-(6) can have strong impacts on the outcome of the TSC test.

In the DUST case, for example, systematically positive $\Delta$RMSD was detected only in one prognostic variable and only over land (cf. Fig. 5c for results at $t = 5\,\mathrm{min}$; results at later time are similar thus not shown). If we had not included aerosol concentrations in the list of monitored variables, or had not chosen to calculate the test diagnostics over land and ocean separately, the TSC test would have given a false "pass" (i.e., a false negative result). While the limited number of test scenarios included in this study have been categorized as expected by the current test setup, there might be more subtle cases, e.g., minor bugs in the code, that require further adjustment of aspects (4)-(6). As a next step, we plan to include a number of bug fixes and additional parameter modifications from the recent model development activities to further evaluate the TSC test setup.

Results in Fig. 4 revealed that $\mathcal{P}_{\mathrm{min},t}$ in passing and failing cases evolve differently. Considering the inherent nonlinearities in the model equations and the resulting variability in the numerical solutions, a pass/fail criterion that characterizes the time series of $\mathcal{P}_{\mathrm{min},t}$ using multiple time steps is expected to provide more accurate test results than a criterion based on one time step. In this paper we made a simple and preliminary choice, requiring all $\mathcal{P}_{\mathrm{min},t}$ diagnosed between $t = 5\,\mathrm{min}$ and $t = 10\,\mathrm{min}$ to fall below a threshold of $0.5\,\%$ in order for a case to fail the test. Adopting a more refined criterion, e.g., one that takes into account not only the magnitude of $\mathcal{P}_{\mathrm{min},t}$ but also its trend, might allow us to further shorten the integration time. The impacts of ensemble size and initialization strategy were not explored in this study, but are worth investigating in future work.

## 5.2 Impact of noisy parameterization

As mentioned in the introduction, the radiation parameterization in CAM5 uses a random number generator that leads to state-dependent noise in the model results. All the simulations presented in Sect. 4 were conducted with a fixed time step size ratio between radiation and the other physics parameterizations, with radiation calculated every other time step. We also conducted TSC simulations with radiation calculated only at the first time step. The impact is illustrated by Fig. 6 where one failing case, CONV-LND, is shown together with two passing cases, Titan-PGI and YS-Intel15-O2. The time series of $\mathcal{P}_{\mathrm{min},t}$ in the CONV-LND case is not distinguishable from the passing cases in the first 3 min of model integration when radiation was called frequently, but already distinguishable after the first minute when radiation was called only once. Substantial decrease of initial $\mathcal{P}_{\mathrm{min},t}$ in the "radiation-once-only" configuration was also seen in several other test scenarios. Our interpretation of this observation is that noise in the model makes it harder to detect signal associated with parameter perturbation, thus requiring longer spin-up in the TSC test. This implies that for models that have very noisy physics, e.g., those with stochastic parameterizations, the TSC simulations might need to be longer than proposed here. Hodyss et al. (2013) demonstrated that noise in a discrete model can result in reduced convergence rate or even loss of convergence. We speculate that the TSC method can still be useful as long as the model has an appreciably positive convergence rate (recall that the time step convergence in CAM5 features a slow rate of 0.4). It will be interesting to explore the utility of our method in models with stochastic parameterizations.

## 5.3 Comparison with other test methods

The development of the TSC test was motivated by the loss of utility of the PERGRO method and the relatively high computational cost of CAM-ECT. Since all three are regression testing methods, it is worth clarifying some linkages and distinctions among them.

CAM-ECT compares the model climate, and considers two sets of results "the same" when ensembles of one-year simulations show consistent statistical distributions of global annual averages. PERGRO and TSC view CAM as a deterministic model, and considers two sets of model results "the same" when the observed solution differences with respected to trusted solutions appear to be consistent with the expected evolution of initial perturbation or time stepping error. In PERGRO and TSC, one-to-one solution comparisons are conducted using instantaneous grid-point values, and the solution differences are evaluated well within the deterministic limit of the flow evolution.

From the perspective that climate is essentially the statistical characterization of deterministic-scale atmospheric conditions, and the fact that the same set of differential-integral equations control the short-term and long-term behaviors of the atmospheric motion in a numerical model, one can expect the different regression testing methods to provide the same "pass" or "fail" results when the solution differences are either very small (e.g., at round-off level) or very different (e.g., due to a major bug in the code). The general consistency between the TSC results shown in this paper and the corresponding test results from Baker et al. (2015) provides evidence to support this reasoning. On the other hand, since the different methods assess the magnitude of solution change with different criteria and at different time scales, we expect there will be cases when they give different answers. The NU case (cf. Table 1 and Sect. 4) that passed CAM-ECT but failed TSC is one such example. As a possible opposite example, we note that within the step size range of 1 s to 1800 s, the time step convergence in CAM5.3 is slow (the rate is about 0.4) and the time step sensitivity is strong (Wan et al., 2015). In other words, in the few-second time step range, the solutions are converging but have not yet converged. For this reason, we speculate that some subtle solution changes might pass the TSC but fail CAM-ECT.

For practical model testing, it is highly desirable to find methods capable of detecting early signs of climate-changing results at low computational cost and with fast test turnaround. However, it is worth noting that the word "climate-changing" is ambiguous until a quantitative criterion is specified. For example, two simulations representing indistinguishable climate characteristics according to SIEVE (cf. Sect. 1) based on the AMWG diagnostics package (https://www2.cesm.ucar.edu/working-groups/amwg/amwg-diagnostics-package) might be distinguishable using additional metrics or using CAM-ECT. Similarly, two simulations determined to be consistent using CAM-ECT based on the global and annual averages might turn out distinguishable using grid-point-wise model output and monthly time series. As for the TSC method, the relatively strong time step sensitivity in CAM5 implies that the numerical accuracies are substantially different when time step size is changed, hence a test procedure based on time step convergence also includes some level of ambiguity. As can be seen in Fig. 2, if we had chosen to conduct a TSC test using a 1800 s time step instead of 2 s, the results from the RH-MIN-HIGH case (which was determined by CAM-ECT as climate-changing) would have been assigned a "pass" by TSC. In the future, if CAM's convergence rate is improved and the accuracy of time stepping increased, one can expect TSC test conducted with 2 s step size to be capable

of detecting more subtle solution differences. Since there are flexibilities in the TSC test (cf. Sect. 5.1), we expect it will be possible to adjust the test setup so that the outcome closely matches the results from CAM-ECT or other methods that compare the model climate with a clearly defined criterion for "climate-changing" results. Future work is planned to further compare TSC with other regression testing methods.

## 6 Conclusions

In this study, we designed and evaluated a test procedure for determining whether the solutions of a numerical model remain the same within the limit of the time integration accuracy when the bit-for-bit reproducibility is lost due to code modifications or computing environment changes. A "fail" signal is issued when the numerical solutions no longer converge to the reference solutions of the original model. The test method is deterministic by nature, but involves an ensemble of simulations to account for possible flow dependencies of the numerical error.

Using the CAM5 model, we provided initial evidence that the test procedure based on $10\,\mathrm{min}$ simulations with $2\,\mathrm{s}$ step size (i.e., a total of 300 time steps per simulation) can be used to distinguish situations where solution differences were deemed insignificant or substantial by a different testing method based on assessment of the simulated climate statistics. The new test is not exhaustive since it does not detect issues associated with diagnostic calculations that do not feedback to the model state variables. Nevertheless it provides a practical, objective and computationally inexpensive way to assess the significance of solution changes. Our experience showed that, using supercomputing facilities, the wall clock time for conducting an ensemble of 12-member simulations typically ranges from a few minutes to a few hours. Such fast turnaround makes the new test a convenient tool for model testing. Future studies are planned to further evaluate the new method using more test scenarios, compare it with other methods of regression testing, and optimize the implementation of the strategy. We also plan to assess the feasibility of applying the test to subcomponents of the model code for the purpose of unit testing and debugging.

The new test is built on the generic concept of time step convergence, and the implementation does not require any code modifications. We plan to explore the utility of the method in other components of our Earth system model (e.g., ocean, sea ice, and land ice), and expect that the same concept is applicable to a wide range of geophysical models such as global and regional weather and climate models, cloud resolving models, large eddy simulations, and even direct numerical simulations.

## 7 Code and data availability

The source code of CAM5 can be obtained as part of the Community Earth System Model (CESM) from the public release website https://www2.cesm.ucar.edu/models/current. The scripts for conducting and analyzing the ensemble simulations, and the simulation data discussed in the paper, are available from the corresponding author upon request.

*Acknowledgements.* The authors thank Dr. W. Sacks (NCAR) and the two anonymous reviewers for their valuable comments and suggestions. This research was supported as part of the Accelerated Climate Modeling for Energy (ACME) program, funded by the U.S. Department of

Energy, Office of Science, Office of Biological and Environmental Research (BER). The basis of the work, the time step convergence study, was previously supported by BER as part of the Scientific Discovery through Advanced Computing (SciDAC) Program, and by the Linus Pauling Distinguished Postdoctoral Fellowship of the Pacific Northwest National Laboratory (PNNL). This research used high-performance computing resources from the Oak Ridge Leadership Computing Facility at the Oak Ridge National Laboratory, supported by the Office of Science of DOE under Contract No. DE-AC05-00OR22725, and the National Center for Atmospheric Research (NCAR) Computational and Information Systems Laboratory, sponsored by the National Science Foundation. PNNL is operated by Battelle Memorial Institute for DOE under contract DE-AC05-76RL01830. NCAR is sponsored by the National Science Foundation.

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

**Table 1.** CAM5 simulations conducted to evaluate the effectiveness of the TSC method. Simulations in group ENV used the same code but different computers, compiler versions, or optimization levels. Group MOD includes code modifications following Milroy et al. (2016). Group PAR includes parameter perturbation simulations following Baker et al. (2015). The pass/fail criterion and the definition of $\mathcal{P}_{\min,t}$ can be found in Sect. 3.2.

| Group | Case name | Computer | Compiler/ optimization | Code change | Model parameters | Pass/fail expected | Pass/fail from TSC | $\mathcal{P}_{\min,t}$ 5–10 min avg. | $\mathcal{P}_{\min,t}$ at $t=5$min |
|---|---|---|---|---|---|---|---|---|---|
| - | CTRL | Titan | Intel 15.0.2 –O2 | No | All default | - | - | - | - |
| ENV | Titan-PGI | Titan | PGI 15.3.0 –O2 | No | All default | Pass | Pass | 11 % | 6.4 % |
| ENV | YS-Intel15-O2 | Yellowstone | Intel 15.0.0 –O2* | No | All default | Pass | Pass | 4.5 % | 3.8 % |
| ENV | YS-Intel15-O3 | Yellowstone | Intel 15.0.0 –O3* | No | All default | Fail | Fail | $3.8\times10^{-12}$ % | $1.0\times10^{-11}$ % |
| MOD | DM | Titan | Intel 15.0.2 –O2 | Yes | All default | Pass | Pass | 8.6 % | 6.2 % |
| MOD | P | Titan | Intel 15.0.2 –O2 | Yes | All default | Unknown | Pass | 7.8 % | 4.2 % |
| PAR | DUST | Titan | Intel 15.0.2 –O2 | No | dust_emis_fact = 0.45 (0.55) | Fail | Fail | $1.6\times10^{-3}$ % | $1.9\times10^{-3}$ % |
| PAR | FACTB | Titan | Intel 15.0.2 –O2 | No | sol_factb_interstitial = 1.0 (0.1) | Fail | Fail | $2.5\times10^{-6}$ % | $8.6\times10^{-6}$ % |
| PAR | FACTIC | Titan | Intel 15.0.2 –O2 | No | sol_factic_interstitial = 1.0 (0.4) | Fail | Fail | $4.8\times10^{-7}$ % | $4.6\times10^{-7}$ % |
| PAR | RH-MIN-LOW | Titan | Intel 15.0.2 –O2 | No | cldfrc_rhminl = 0.85 (0.8975) | Fail | Fail | $3.6\times10^{-15}$ % | $3.5\times10^{-15}$ % |
| PAR | RH-MIN-HIGH | Titan | Intel 15.0.2 –O2 | No | cldfrc_rhminh = 0.9 (0.8) | Fail | Fail | $9.2\times10^{-14}$ % | $3.3\times10^{-14}$ % |
| PAR | CLDFRC-DP | Titan | Intel 15.0.2 –O2 | No | cldfrc_dp1 = 0.14 (0.10) | Fail | Fail | $2.1\times10^{-9}$ % | $4.0\times10^{-9}$ % |
| PAR | UW-SH | Titan | Intel 15.0.2 –O2 | No | uwschu_rpen = 10.0 (5.0) | Fail | Fail | $2.0\times10^{-9}$ % | $3.7\times10^{-9}$ % |
| PAR | CONV-LND | Titan | Intel 15.0.2 –O2 | No | zmconv_c0_lnd = 0.0035 (0.0059) | Fail | Fail | $9.0\times10^{-4}$ % | $4.7\times10^{-3}$ % |
| PAR | CONV-OCN | Titan | Intel 15.0.2 –O2 | No | zmconv_c0_ocn = 0.0035 (0.045) | Fail | Fail | $6.7\times10^{-10}$ % | $8.1\times10^{-10}$ % |
| PAR | NU-P | Titan | Intel 15.0.2 –O2 | No | nu_p = $1.0\times10^{14}$ ($1.0\times10^{15}$) | Fail | Fail | $2.5\times10^{-10}$ % | $1.4\times10^{-10}$ % |
| PAR | NU | Titan | Intel 15.0.2 –O2 | No | nu = $9.0\times10^{14}$ ($1.0\times10^{15}$) | Fail | Fail | $1.4\times10^{-5}$ % | $1.5\times10^{-5}$ % |

* Model was compiled without the "-fp-model" flag; All the other Intel simulations in the table used "-fp-model source" for Fortran and "-fp-model precise" for the C code.

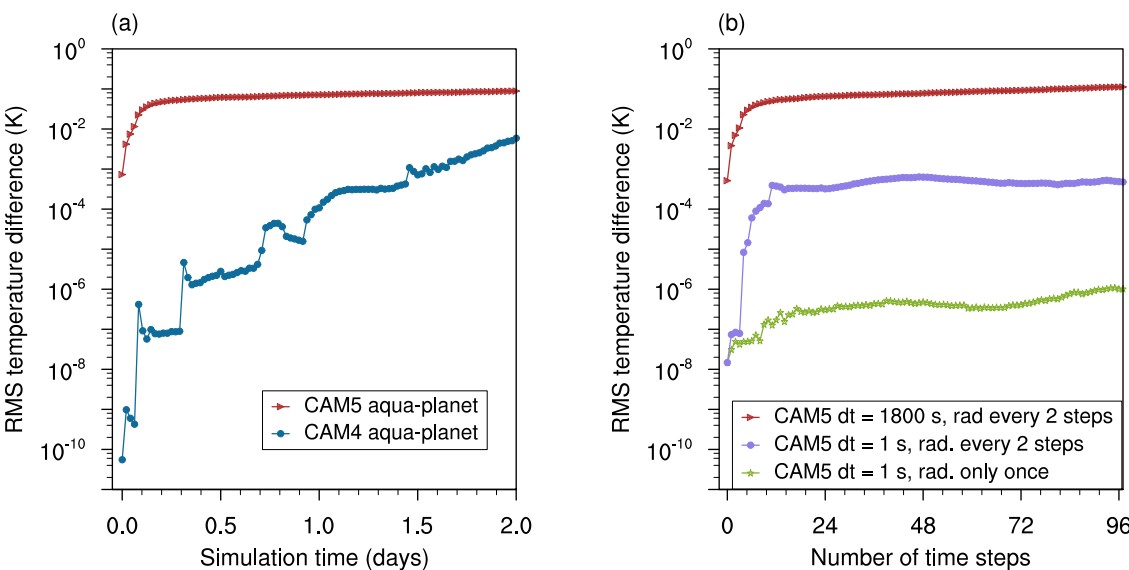

**Figure 1.** Examples of the evolution of RMS temperature difference (unit: K) caused by random perturbations of order $10^{-14}$ K imposed on the temperature initial conditions. (a) Aqua-planet simulations conducted with the CAM4 (blue) and CAM5.3 (red) physics parameterization suites using the default 1800 s time step. (b) Simulations conducted with the CAM5.3 physics suite using the default 1800 s time step and with radiation calculated every other step (red), using 1 s time step and with radiation calculated every other step (purple), and using 1 s time step and with radiation calculated only once at the beginning of the integration. All simulations used the spectral element dynamical core at approximately 1° horizontal resolution.

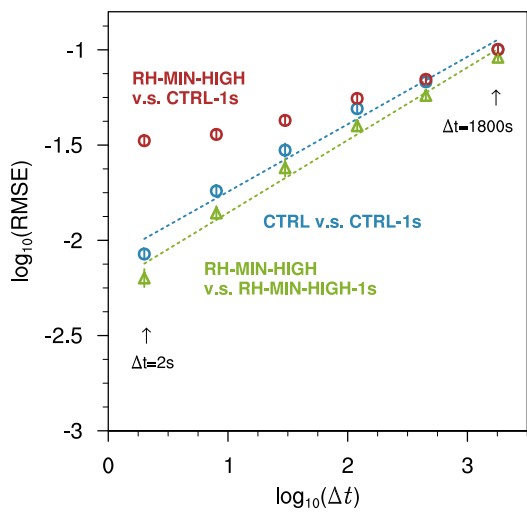

**Figure 2.** Convergence diagram showing the RMS solution differences calculated using the instantaneous 3D temperature field after $1\,\mathrm{h}$ of CAM5 integration. Blue circles and green triangles are the RMS differences relative to reference solutions obtained with the same code but using a $1\,\mathrm{s}$ time step. Red circles are the RMS differences between the reference solution of the CTRL model ($1\,\mathrm{s}$ time step) and the RH-MIN-HIGH simulations with longer step sizes. Each marker shows the average RMS difference of 12 ensemble simulations that used different initial conditions sampled from different months of the year; the bars indicate the $\pm\sigma$ ranges where $\sigma$ denotes the ensemble standard deviation. The dashed lines are linear fits between $\log_{10}(\mathrm{RMSD})$ and $\log_{10}(\Delta t)$.

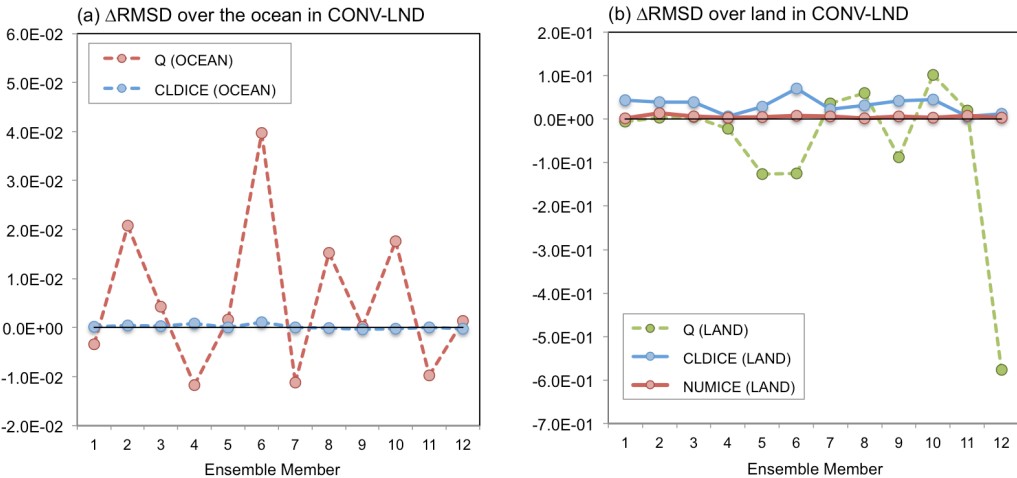

**Figure 3.** $\Delta\mathrm{RMSD}_{t,j,m}$ of individual ensemble members after $t = 5\,\mathrm{min}$ of model integration in the "CONV-LND" test case that was designed to fail the TSC test when all variables, domains, and ensemble members are considered (cf. Table 1 and Sect. 4.2). The values have been normalized by the mean RMSD of the trusted ensemble, i.e., $\overline{\mathrm{RMSD}}_{\mathrm{trusted},t,j}$, of the corresponding prognostic variables and domains. (a) ocean; (b) land. Dashed (solid) lines correspond to variables that passed (failed) the TSC test according to the criterion defined by Eq. (6). The prognostic variables shown in the figure are specific humidity (Q), grid-box mean ice crystal mass concentration in stratiform clouds (CLDICE), and grid-box mean ice crystal number concentration in stratiform clouds (NUMICE).

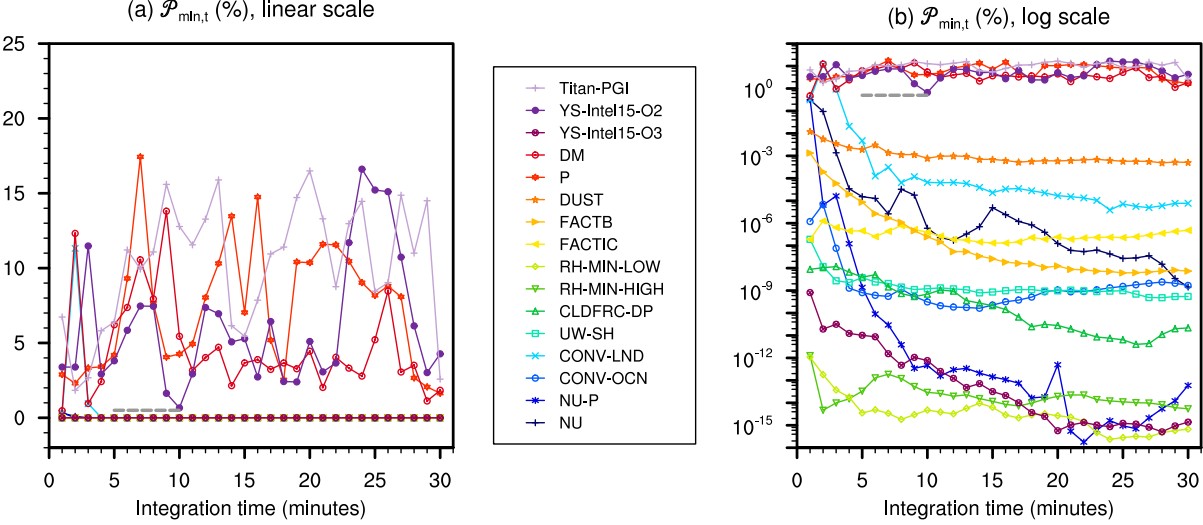

**Figure 4.** $\mathcal{P}_{\min,t}$ as a function of model integration time, plotted in linear scale (a) and in logarithmic scale (b). The dashed gray lines indicate the threshold for assigning an overall "pass" or "fail" to a test ensemble (cf. Eq. 8 and the text above it).

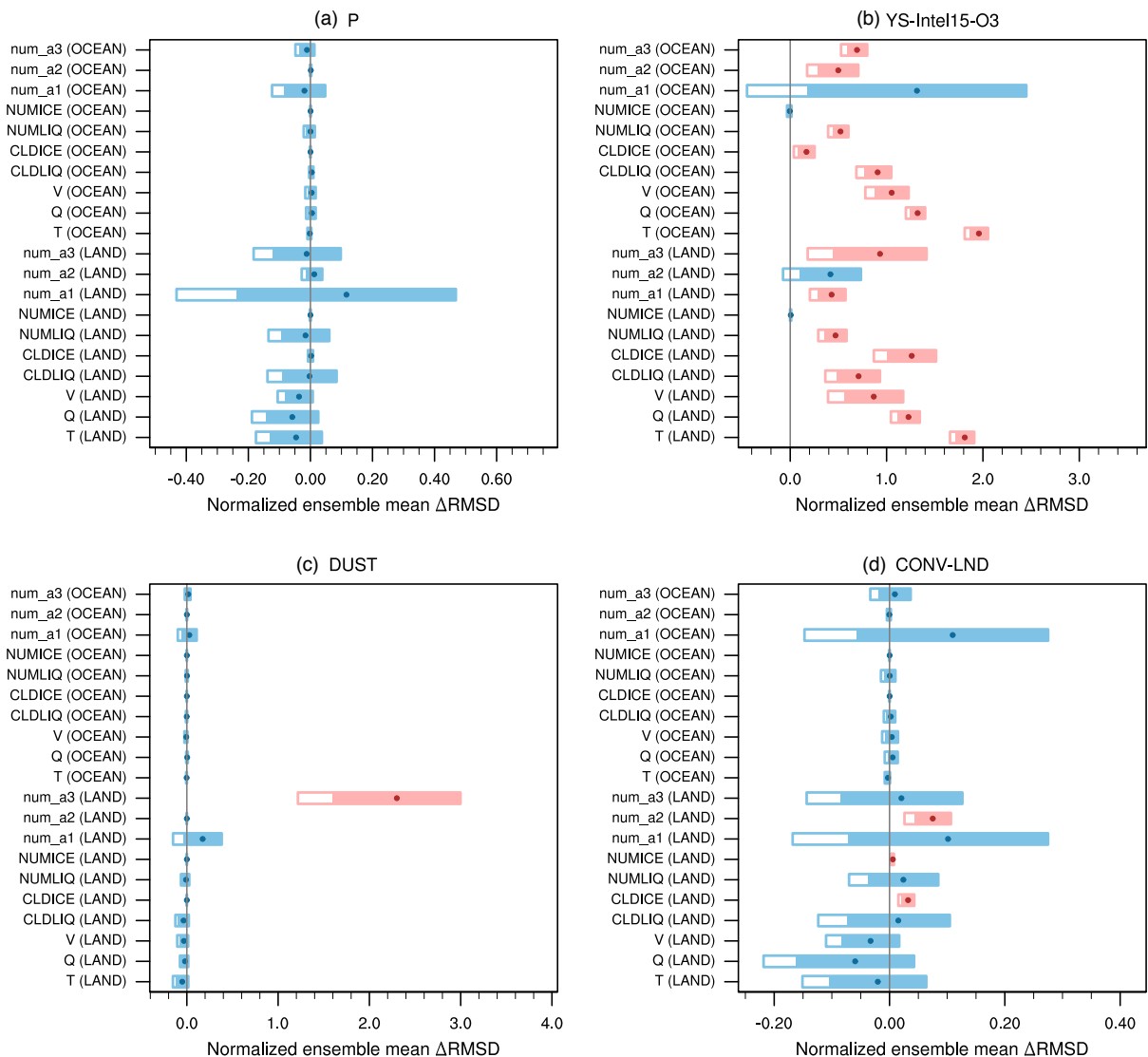

**Figure 5.** Ensemble mean $\overline{\Delta\mathrm{RMSD}}_{t,j}$ (dots) and the $\pm 2\sigma$ range of the mean (filled boxes) where $\sigma$ denotes the standard deviation. The left end of an unfilled box shows the threshold value corresponding to $\mathcal{P}_0 = 0.5\,\%$ in the one-sided $t$-test. All values shown here have been normalized by the mean RMSD of the trusted ensemble, i.e., $\overline{\mathrm{RMSD}}_{\mathrm{trusted},t,j}$, of the corresponding prognostic variable and domain (cf. y-axis labels). Red and blue indicate fail and pass, respectively, according to the criterion defined by Eq. (6). Results are shown at $t = 5\,\mathrm{min}$ for four test cases: (a) P, (b) YS-Intel15-O3, (c) DUST, and (d) CONV-LND. The test case configurations are explained in Table 1 and Sect. 4.

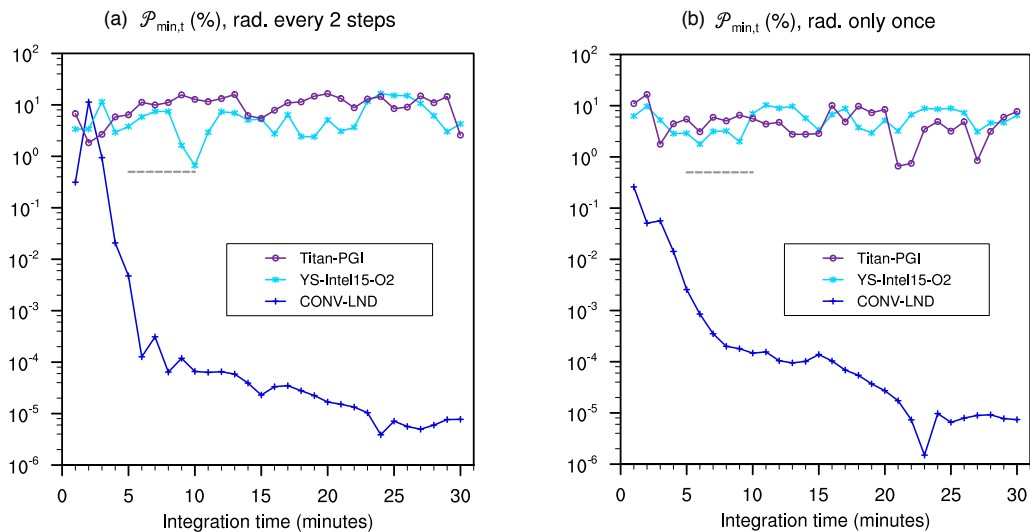

**Figure 6.** As in Fig. 4b, but showing only a few test scenarios to compare the results obtained from simulations where (a) radiation is calculated every other time step, and (b) radiation is calculated only at the beginning of the integration.