# Peer review of "A new and inexpensive non-bit-for-bit solution reproducibility test based on time step convergence (TSC1.0)"

_Geoscientific Model Development, 2016_

## Short Comment (SC1) · 30 Jun 2016

This is a clever idea, and the paper is very well written.

I'd like to be convinced that this technique truly has more power than seemingly simpler techniques. For example, can some of the same experiments be redone with this set of runs?:

(1) control: unmodified model with 1s time step

(2) baseline for comparison: unmodified model with 1s time step, with a roundoff-level perturbation in the temperature field

(3) test code: some change in the code with 1s time step

Basically, I'd like to be convinced that the "time step convergence" is truly needed here, and that it truly provides more power than just comparing two versions of the model with a short time step. Does the above, conceptually simpler test give false positives or false negatives in cases where the TSC test gives the correct answer?

I'd also like clarification on the following point: On a continuum from non-answer-changing to answer-changing, I see mention of the following types of changes: (1) bit-for-bit identical, (2) answer-changing only at the round-off level, (3) answer-changing only within the limits of numerical accuracy due to the discrete time step size, and (4) climate changing, according to criteria like SIEVE or CAM-ECT. The TSC test distinguishes changes at level 3 or lower from those at level 4. But is there actually a level in between (3) and (4): changes that affect the model evolution in an appreciable way, but are not large enough to cause statistically detectable changes in climate? It seems that many bugs might fall into this intermediate regime – e.g., accidentally flipping the sign on a minor term in an equation. Do the authors feel that there is a set of changes that falls between (3) and (4), and if so, how do they expect these changes to be categorized by the TSC test?

---

## Referee Comment (RC1) · Anonymous Referee #1 · 28 Jul 2016

———————— General comments:

Overall the paper was well-written and clear. The TSC test idea is a clever application of the time step convergence work from Wan et al. (JAMES, 2015) and appears useful. Certainly this approach is promising and inexpensive, and the manuscript is a good start. More details on the manuscript are provided below, but my main concerns to address are as follows:

(1) The paper would have been stronger if the test parameters had been fleshed out more thoroughly, particularly the ensemble size, the false positive rate, and number of variables. For example, because this test returns a "fail" if a single variable fails, then a larger subset of variables will increase the possibility of failure by chance, so making

the reader aware of this relationship would be useful.

(2) More details on the scope of the test would be helpful. There are bits in section 2.1 and later in 4, but it would help to better quantify the scope beyond the "equation-solving" part. In particular, the selection of variables would seem to impact the scope. Because of the limited (?) scope, an example of a bug/issue that is not caught would be helpful. And ideally this counter-example would be discussed within a larger discussion of scope as relating to the choice of variables.

(3) The experimental results section would be stronger if the experiments more closely represented the stated scope (see previous comment). Then the reader would gain a better understanding of the tool's utility. The chosen experiments are essentially the same as those in Baker, et al. (GMD, 2015). While it is important to include those, it is not clear that the 10 variables chosen would be sufficient to catch errors in all parts of the code (as stated in section 2.1), so it would be helpful to have an example of an error that is not caught. Also, several times (e.g., section 2.1) "code modifications" are mentioned as an application for this test, but there is not an example supporting this statement (and including such an example seems important).

(4) Regarding the TSC's use of the t-test, please clarify the reason for the directional t-test. In particular, why does the test only check if the mean is larger than zero? (i.e., $mu\_j > 0$) - as opposed to the non-directed alternative hypothesis: $m\_j \neq 0$. Certainly $mu\_j$ can be negative, so is this scenario just not of concern? For example, in figure 3, if delta_RMSD for variable was negative for all 12 members, then TSC would issue a pass. I am not necessarily questioning the efficacy of the test procedure, but I have to wonder if systematically negative results can be problematic as well or even indicative of an issue with the simulation being tested.

——————— Specific comments:

(1) Section 1: line 20: Check the use of "reproducibility" in this context.

(2) Section 1: The first couple paragraphs are quite similar in parts to the text in Baker, et al. (GMD, 2015), including the same references and some of the same phrases, which is a bit awkward.

(3) page 3, line 24: The tool's application to "code modifications" is mentioned here and in section 5, but I don't believe this is being tested in the experiments. It may be of interest to look at CESM code modification experiments in the followup to Baker, et al. (GMD, 2015), which is:

Daniel J. Milroy, Allison H. Baker, Dorit M. Hammerling, John M. Dennis, Sheri A. Mickelson, and Elizabeth R. Jessup, "Towards characterizing the variability of statistically consistent Community Earth System Model simulations." Procedia Computer Science (ICCS 2016), Vol. 80, 2016, pp. 1589-1600. (http://www.sciencedirect.com/science/article/pii/S1877050916309759)

(4) Section 2.1: I would really like to better understand how the selection of the 10 variables affects (or does not affect) the scope.

(5) page 2, line 25: This is not exactly true as CAM-ECT has been used to pinpoint errors in specific code modules (e.g. FMA error on Mira detailed in Milroy et al. 2016).

(6) page 3, lines 27-28: Regarding "...when the accuracy limits related to the algorithmic implementation are taken into account." This doesn't appear to be considered in the rest of the paper.

(7) page 4, line 14: I agree with #5 as a desirable feature, but I don't believe that evidence was given in this manuscript that TSC fulfills #5. Certainly no evidence was given in Baker, et al. (GMD, 2015) that CAM-ECT satisfies #5, though one can imagine the framework could possibly apply. So if the claim is that TSC fulfills this while CAM-ECT does not, it would be stronger to provide specific evidence of such a case for TSC (i.e., an experiment to validate the claim).

(8) Section 2.3: Since the starting conditions for the TSC ensemble are samples from

"a previously conducted long-term simulation", does one need to update this simulation with answer-changing CESM tags, for example? Also does "long-term" mean 1-year or ????? Please give more details on how this part of the process works.

(9) Would the TSC test results be affected if the ensemble was created instead by perturbing initial conditions (since this does not require a previous simulation)?

(10) page 5, last paragraph: Should point out that this test (RH-MIN-HIGH from .8 to .9) is from Baker, et al. (GMD, 2015) for comparison.

(11) page 5, line 34-page 6, line 1: "[...] concept of self-convergence since no structural changes [...] have been introduced into the model." More generally (and relevant to the discussion in Sect 3.2), what if the modified model's 2s timestep behavior is closer to the 1s timestep reference model than to itself for 1s timestep? In other words, what if its convergence behavior to the reference model is different than its self-convergence?

(12) page 6, line 13: The "more substantially" comment is a bit vague. The change is already labeled "climate-changing", which itself seems substantial. Certainly this change is more substantial than, for example, changing the order of operations in the code or something similarly "minor". Clarify?

(13) page 7, first paragraph: Did the authors use the SIEVE method to verify all of the results presented? It is not clear. Also wondering if the example (NU) in Baker, et al. (GMD, 2015) that passed (but that Baker et al. claim should have failed) was independently verified by the authors with SIEVE?

(14) Followup to (12): Recommend that the authors come up with another example of a small scale change that CAM-ECT would not catch because of its use of the global and annual mean (but that TSC would) - other than the NU test from Baker et al. (GMD, 2015). This would probably have to be more subtle than the experiments in Baker, et al. (GMD, 2015). I think this recommendation is particularly pertinent given the list of desired features on page 2 (and that TSC should achieve #6 while CAM-ECT will not).

(15) page 7, line 16: A false negative example would be a great addition and improve the reader's understanding of the tool's scope.

(16) Section 3.2: The splitting into the two domains could be explained more (it is discussed a bit again later in 4.1). It seems a bit arbitrary and suggests that the DUST and CONV-LND failures cannot be detected otherwise. One issue is that by splitting into domains (effectively doubling the number of variables), the false positive rate is being increased. Would be helpful to have more guidance on variable selection and limitations.

(17) page 8, lines 16-28: I'm still struggling a bit with understanding the scope, which is discussed again here in terms of what will and won't be caught (e.g., aerosol concentrations). Please clarify earlier and consider including supporting experimental results.

(18) page 9, line 10: If the implicit assumption that the random variables (mu-sub-j) are Gaussian distributed is violated, will the TSC test results be affected? (And has this been explored? An example could be something like truncation...)

(19) page 9, Step 3 (line 30-> page 6): More clarification is needed here. For the t-test, the choice of .05% is conservative (as acknowledged in text), and it is clear that the specified t-statistic (4.437) is dependent on both the .05% cutoff *and* the sample (ensemble) size (M=12). However, there is a less intuitive dependence on the number of variables that should be pointed out (and discussed). Because the t-test is performed on each variable *individually*, then the number of variables examined certainly affects the overall test failure rates. The conservative choice of .05% may make sense for the 20 variable subset (meaning that a single variable has to fail quite badly to cause a failure of the overall test). However, if one were to use 2 variables (or 100 variables), the .05% may no longer be the best choice. I think this should be addressed given the discussion on page 8 (line 25) that one could choose to include more (and presumably fewer) fields.

(20) page 9, line 9): Please clarify the reason for the directional t-test and consider

updating/clarifying the accompanying discussion on page 9,line 25 -> page 10, line 4.

(21) page 9, line 10: A minor point, but technically one cannot "accept" the null hypothesis. (One can fail to reject the null hypothesis or reject it.)

(22) page 11, line 1: Was the .89 vs. .897 detectable by SIEVE? Also how long of a simulation was run for SIEVE in this case?

(23) page 10: Given the FMA issues found for Mira in Milroy et al. 2016 (and also for BlueWaters), I am questioning the Cori results a bit - also because the results in Table 1 for Cori are not as definitive as for the other machines. Cori uses FMA by default, or was it disabled for these experiments? How long were the simulations examined by SIEVE for Cori?

(24) page 13, line 25: " ...failing the test will very likely mean the climate will be different. Passing the convergence test should hence be considered a necessary condition..." I don't quite agree with this. Many of the parameterizations could be quite different in the short term (because of sensitivity), but the longer term behavior is basically the same. In other words, the weather after 150s may look different (e.g., raining or not), but the annual climate is the same. (This assertion is also made on page 7, lines 15-16)

———————— Technical Corrections

(1) page 2, line 3: Remove the final word "did" from the sentence.

(2) page 2, line 29: The second occurence of "simulation" should be plural.

(3) page 3, line 9: Spell out the number 3 (three).

(4) page 3, lines 23-25: Consider breaking this sentence into smaller parts.

(5) page 5, line 8: "thus saves" should be "thus saving".

(6) page 5, line 18: "dependences" should be "dependencies".

———————— Final thoughts

Interactive
comment

[Figure]

I like the idea of this work, and I hope that the comments and suggestions provided will be helpful for the revision of the paper. I believe that more flushed out algorithm details, a clarification of scope, and better alignment of the experimental results with the stated features of the test will strengthen the paper and its impact and utility.

---

## Referee Comment (RC2) · Anonymous Referee #2 · 4 Aug 2016

Apologies for being so late with my initial comments.

Agree with other reviewers that the paper is overall well-written and clear. I do have some questions and concerns, which are outlined below.

o In the test scenario given the drastically shortened simulation length (5 minutes) with much shorter time steps (1 or 2 seconds), how often are the physical parameterizations (radiation and non-radiation physics) executed? Is it only once for the entire run? If only once, is this a weakness in the overall test design?

o Are all of the outputs from the physical parameterizations that are used in the dynamics applied as tendencies rather than adjustments? Presumably yes, since the

effects of any parameterization that applies its effects as a hard adjustment will not be mitigated by a much shorter time step.

o Is it true that the very rapid growth of a perturbation is due entirely to the physical parameterizations rather than the dynamics? If so, it would be good to point this out specifically, meaning that more traditional means of code verification could still be applied for changes to the dynamical core, assuming the ability to run the model adiabatically.

o Page 2, #50: Regarding the PerGro test using CAM4, presumably the test always fails due to Condition 1 from Rosinski and Williamson (1997): "During the first few time steps, differences between the original and ported code solutions should be within one to two orders of magnitude of machine rounding". If this is correct, it would help to clarify as the primary reason for failure.

o Page 2, #55: It is stated that "Recent versions of the model have become so complicated that rounding level differences in the initial condition can result in very rapid divergence of the simulations". It is not obvious, and no evidence is presented, that code "complication" is a reason for the faster growth. Is it possible, for example, that the initial condition has points which lie on a code branch ("if" test)? Or more generally, perhaps the new physics is driving some quantity such as temperature toward a value which lies on a branch, such as the freezing point of water? If implemented via a tendency equation, the computed value may be one mantissa bit greater than, or one mantissa bit less than, the actual freezing point of water. If a subsequent "if" test applies substantially different algorithms across "true" and "false" branches of a test versus the freezing point, this can be a reason for rapid growth not necessarily related to code complication. This exact scenario was encountered many years ago when testing growth behavior with the relatively simple BATS land model in CAM.

o Page 3, #65: It is stated that "The very fast evolution of initial perturbation is caused by multiple factors". What are those factors? Similar to the previous point, a weakness

of the paper is that it does not describe any of the reasons for rapid growth. There is only speculation that code complication is to blame.

o Page 5, #125: Generally commutative operations are not answer-changing. Instead perhaps the authors mean "associative operations"?

o Page 6, #170: How is the convergence rate of 0.4 calculated?

o Page 9, #285: Definition of the two separate domains is presumably land and ocean. It would help readability to state this up front, and also the reasons for the choice.

o Page 15, #495: If passing the test doesn't guarantee that the model will produce the same climate characteristics, isn't this a weakness of the procedure? I thought the main point of the procedure was to provide a mechanism to enable non-experts to confidently commit roundoff-level code changes to the repository.

The "major revisions" requested involve a much more thorough analysis of the reasons for rapid perturbation growth in CAM4 and CAM5. Speculation about "code complexity" is not adequate. The example cited by this reviewer of rapid growth caused by a simple land scheme (BATS) was really a bug not a feature of the scheme. It would be nice to have some assurance that this possibility (ill-formed or buggy algorithms) has been explored to some extent with the current CAM model.

———————————————

---

## Author Comment (AC1) · 21 Oct 2016

We thank the referee for the insightful comments and suggestions. Our responses are detailed below.

**Comment:** *Apologies for being so late with my initial comments. Agree with other reviewers that the paper is overall well-written and clear. I do have some questions and concerns, which are outlined below.*

*In the test scenario given the drastically shortened simulation length (5 minutes) with much shorter time steps (1 or 2 seconds), how often are the physical parameterizations (radiation and non-radiation physics) executed? Is it only once for the entire run? If only*

[Figure]

*once, is this a weakness in the overall test design?*

**Response**: Simulations presented in the discussion paper had all parameterizations calculated every time step except for radiation which was called only once. We have repeated the simulations with radiation calculated every other time step (i.e., using the same time step ratio between radiation and the other parameterization as in the default model). We found that the TSC results were similar to those in the discussion paper in the sense that the simulations that were expected to "pass" showed typical $\mathcal{P}_{\min}$ values between a few percent and $\sim$20% during a model time of 30 minutes, while those expected to "fail" showed $\mathcal{P}_{\min}$ values substantially smaller than 1% after a short (few-minute) spin-up.

It is worth noting that radiation is the only part in the current atmosphere model code that contains intentionally introduced randomness at magnitudes way beyond the level of rounding error. The radiation code uses a pseudo random number generator, and the seeds for the random number generator are chosen from the least significant digits of the pressure field. This effectively introduces state-dependent noise to the numerical solution, and is one of the reasons for the very rapid growth of initial perturbation (see also our response to respective comments below). In the revised manuscript, we present both sets of simulations (i.e., with radiation called at every other time step or only once), and include a discussion on the impact of noise on the utility of the TSC method.

**Comment:** *Are all of the outputs from the physical parameterizations that are used in the dynamics applied as tendencies rather than adjustments? Presumably yes, since the effects of any parameterization that applies its effects as a hard adjustment will not be mitigated by a much shorter time step.*

**Response**: Yes, in the version of CAM5 we used in this study, the impacts of the parameterized physics are provided as tendencies to the dynamical core. Within the physics parameterization suite, however, processes are calculated with sequential
splitting meaning that the tendencies from one parameterization are used to update the model state variables before those variables are passed onto the next parameterization. The sequential splitting still causes large time integration error when used in combination with long time steps (as is the case in CAM5 which uses a 30-minute time step for the coupling between different parameterizations and between physics and dynamics), because the splitting allows individual processes to operate in isolation for a long time (i.e., one time step) without considering the possible interactions between different processes.

**Comment:** *Is it true that the very rapid growth of a perturbation is due entirely to the physical parameterizations rather than the dynamics? If so, it would be good to point this out specifically, meaning that more traditional means of code verification could still be applied for changes to the dynamical core, assuming the ability to run the model adiabatically.*

**Response**: Yes, we clarify in the revised manuscript that the rapid growth is indeed due to the physics parameterizations. Perturbation growth test performed with the spectral transform dynamical core indicated RMS temperature difference on the order of $\mathcal{O}(10^{-12})$ by the end of the second model day. We have not done many simulations with the dynamical-core-only configuration, but given such small magnitudes of RMS temperature difference and the rather slow growth, we expect that the original test strategy is still applicable to and useful for testing of the dynamical core.

**Comment:** *Page 2, #50: Regarding the PerGro test using CAM4, presumably the test always fails due to Condition 1 from Rosinski and Williamson (1997): "During the first few time steps, differences between the original and ported code solutions should be within one to two orders of magnitude of machine rounding". If this is correct, it would help to clarify as the primary reason for failure.*

**Response**: The respective sentences in the discussion paper were: "When the test was originally developed, the physical parameterizations were quite simple, and the

test was robust. The method gradually became less useful as the model became more comprehensive and complex, and compromises were made to preserve some utility for the test.For example, in CAM4, the PerGro test needed to be performed in an aqua-planet configuration, i.e., without the land surface parameterizations, and with a few (small) pieces of code in the atmospheric physics parameterizations switched off or revised, because those codes were known to be very sensitive to small perturbations, and would always lead the test to fail."

We provide the following clarification in the revised manuscript: Rosinski and Williamson (1997) established two conditions for the validation of a ported code:

- Condition 1. During the first few time steps, differences between the original and ported code solutions should be within one to two orders of magnitude of machine rounding.

- Condition 2. During the first few days, growth of the difference between the original and ported code solutions should not exceed the growth of an initial perturbation introduced into the lowest-order bits of the original code solution.

It is important to note that in order for those two conditions to be useful for the intended verification, the model code has to satisfy a "Condition 0":

- Condition 0. During the first few time steps, rounding-level initial perturbations introduced to the original code in the original environment should not trigger solution differences larger than one to two orders of magnitude of machine rounding.

If Condition 0 is violated, it is expected that the ported code will always fail Condition 1 whether there is a porting error or not; in addition, the very rapid growth of perturbations even in a trusted computing environment could make it difficult to distinguish differences between trusted solutions from differences between a trusted solution and

a problematic test solution, causing misleading fulfillment of condition 2. Therefore, if Condition 0 is violated, Conditions 1 and 2 might no longer be useful for porting verification.

When the PerGro test was originally developed, the physical parameterizations were quite simple, the code was able to satisfy Condition 0, and the test method was robust. As the model became more comprehensive and complex, more rapid growth of rounding-level initial perturbation was observed. Compromises were made to preserve some utility for the PerGro test. For example, in CAM4, the test needed to be performed in an aqua-planet configuration, i.e., without the land surface parameterizations, and with a few (small) pieces of code in the atmospheric physics parameterizations switched off or revised, because those codes were known to be very sensitive to small perturbations. If those pieces of codes were not switched off or revised, perturbations on the trusted machine would grow so rapidly that the RMS differences grew to $\mathcal{O}(0.1)$ over a few timesteps. Disabling the land interactions and a few pieces of code returned the bulk of the atmospheric model to a configuration where differences between perturbed and unperturbed initial conditions grew substantially more slowly. Most of the time, the RMS differences grew at a rate well below one order of magnitude per timestep in a trusted environment. An example is shown by the blue curve in Fig. 1 of the discussion paper (see also Fig. 1a in this document). With the revised aqua-planet configuration of CAM4, it was still possible to examine solution differences between original and test solutions to see whether they violated Condition 2 for a port validation effort. But with CAM5, initial perturbations grow too rapidly even in an aqua-planet simulation (see red curve in Fig. 1a below), making the original PerGro method no longer useful for porting test.

**Comment:** *Page 2, #55: It is stated that "Recent versions of the model have become so complicated that rounding level differences in the initial condition can result in very rapid divergence of the simulations". It is not obvious, and no evidence is presented, that code "complication" is a reason for the faster growth. Is it possible, for example,*

*that the initial condition has points which lie on a code branch ("if" test)? Or more generally, perhaps the new physics is driving some quantity such as temperature toward a value which lies on a branch, such as the freezing point of water? If implemented via a tendency equation, the computed value may be one mantissa bit greater than, or one mantissa bit less than, the actual freezing point of water. If a subsequent "if" test applies substantially different algorithms across "true" and "false" branches of a test versus the freezing point, this can be a reason for rapid growth not necessarily related to code complication. This exact scenario was encountered many years ago when testing growth behavior with the relatively simple BATS land model in CAM.*

*Page 3, #65: It is stated that "The very fast evolution of initial perturbation is caused by multiple factors". What are those factors? Similar to the previous point, a weakness of the paper is that it does not describe any of the reasons for rapid growth. There is only speculation that code complication is to blame.*

**Response**: So far we have found three major contributors to the rapid divergence of solutions in the current model:

First, the default time step of 1800 s in CAM5 is significant compared to the characteristic time scales of many physical processes represented by the model, so the increments in the model state (the process tendencies times the model time step) are significant, and the differences between a pair of simulations with slightly different initial conditions can also be perceptible. The red and purple curves in Fig. 1b below show that when the time step sizes of all model components are changed by a factor of 1800, the solution differences after the same number of time steps also change by a similar ratio.

Second, the solar and terrestrial radiation parameterization in CAM5 uses a pseudo random number generator, and the seeds for the generator are chosen from the least significant digits of the pressure field. This effectively introduces state-dependent noise into the numerical solution. The green curve in Fig. 1b below shows the differences

between a pair of simulations conducted with 1 s time step but with radiation calculated only once at the beginning of the integration. Compared to the purple curve where radiation was calculated every other time step, the solution differences were further reduced by about 3 orders of magnitude. We note that the noisiness from the radiation calculation can be controlled by making the random seeds independent of the model state so that the random series become reproducible from one simulation to another. But the radiation example also implies that models with state-dependent stochastic parameterizations might feature rapid perturbation growth as well.

The third reason for rapid perturbation growth has to do with poorly conditioned pieces of code. Two types of examples where discussed by Rosinski and Williamson (1997): (i) an upshift in digit of solution error resulting from division by a small number, and (ii) if-statements associated with algorithmic discontinuity. We have experienced both types of situations in the CAM5 code, although the specific formulae were different from those given in the paper of Rosinski and Williamson (1997). Compared to its predecessors, CAM5 uses modern parameterizations with substantially more detailed description of the atmospheric phenomena, and the model also carries an expanded list of tracers. The increase in model complexity and the corresponding growth in the size of the code substantially increase the chance for poor conditioning to occur.

The explanations above are included in the revised manuscript. We think a more detailed description of our findings is out of the scope of the present manuscript. A separate paper is in preparation:

Singh B., Rasch, P. J., Wan, H., and Edwards, J.: A verification strategy for atmospheric model codes using initial condition perturbations. To be submitted.

**Comment:** *Page 5, #125: Generally commutative operations are not answer-changing. Instead perhaps the authors mean "associative operations"?*

**Response**: Thanks for pointing out this error. We indeed meant "associative". This is corrected in the revised manuscript.

**Comment:** *Page 6, #170: How is the convergence rate of 0.4 calculated?*

**Response**: The convergence rate is the regression coefficient of the linear regression between ensemble mean $\log_{10}$ RMSD and $\log_{10} \Delta t$. This is clarified in the revised manuscript.

**Comment:** *Page 9, #285: Definition of the two separate domains is presumably land and ocean. It would help readability to state this up front, and also the reasons for the choice.*

**Response:** We clarify the following in the revised manuscript: The essence of our new test method is to distinguish solution differences caused by code modifications or computing environment changes from solution differences caused by model time step change (2 s versus 1 s). While certain changes in the model code, e.g., those related to dust emission or convection over land, only affect a limited number of grid points during simulations that are just a few minutes to a hour in length, time step size affects the solution from the first step and at all grid points. Consequently, subtle but "real" solution changes might be masked by the model's time stepping error thus difficult to detect. To help address this challenge, we calculate the test diagnostics for $N_{\text{dom}}$ = 2 domains, i.e., land and ocean. This is a practical and somewhat arbitrary choice that aims at increasing the sensitivity of the TSC test.

**Comment:** *Page 15, #495: If passing the test doesn't guarantee that the model will produce the same climate characteristics, isn't this a weakness of the procedure? I thought the main point of the procedure was to provide a mechanism to enable non-experts to confidently commit roundoff-level code changes to the repository.*

**Response**: Strictly speaking, the TSC test is a method for assessing whether solution differences seen in very short (few-minute) simulations exceed the numerical accuracy of the model's time stepping algorithms. This neither assesses whether the solution differences are at rounding level, nor determines whether the climate characteristics are the same. We note that when PerGro was considered a useful porting validation

method, passing that test did not guarantee the model would produce the same climate, either. Given the invalidity of the PerGro method in CAM5, and the high computational costs associated with conducting and evaluating climate simulations, the TSC method provides a practical and useful alternative to determine whether the model is behaving as expected in the sense that the numerical solutions feature the same time stepping error when compared to a predefined set of reference solutions. This is clarified in the revised manuscript.

**Comment:** *The "major revisions" requested involve a much more thorough analysis of the reasons for rapid perturbation growth in CAM4 and CAM5. Speculation about "code complexity" is not adequate. The example cited by this reviewer of rapid growth caused by a simple land scheme (BATS) was really a bug not a feature of the scheme. It would be nice to have some assurance that this possibility (ill-formed or buggy algorithms) has been explored to some extent with the current CAM model.*

**Response**: We agree with the referee that the reasons for rapid perturbation growth in CAM is an important (and also very interesting) research topic. As mentioned above, we have managed to understand at least some of the causes, and included brief explanations in the revised manuscript. To us, the rapid perturbation growth is a motivation for developing a new test method but not the focus of this manuscript. We will report in detail our findings regarding perturbation growth in a separate paper.
* * *
[Figure]

[Figure]

Figure 1: Examples of the evolution of RMS temperature difference (unit: K) caused by random perturbations of order $10^{-14}$ K imposed on the temperature initial conditions. (a) Aqua-planet simulations conducted with the CAM4 (blue) and CAM5.3 (red) physics parameterization suites using the default 1800 s time step. (b) Simulations conducted with the CAM5.3 physics suite using the default 1800 s time step and with radiation calculated every other step (red), using 1 s time step and with radiation calculated every other step (purple), and using 1 s time step and with radiation calculated only once at the beginning of the integration (green). All simulations used the spectral element dynamical core at approximately 1° horizontal resolution.

**Fig. 1.** Examples of the evolution of RMS temperature difference caused by initial perturbation.

---

## Author Comment (AC2) · 21 Oct 2016

We thank Dr. Sacks for his insightful questions. Our responses are detailed below.

**Comment:** *This is a clever idea, and the paper is very well written. I'd like to be convinced that this technique truly has more power than seemingly simpler techniques. For example, can some of the same experiments be redone with this set of runs?:*
*(1) control: unmodified model with 1s time step*
*(2) baseline for comparison: unmodified model with 1s time step, with a roundoff-level perturbation in the temperature field*
*(3) test code: some change in the code with 1s time step*
*Basically, I'd like to be convinced that the "time step convergence" is truly needed here,*

*and that it truly provides more power than just comparing two versions of the model with a short time step. Does the above, conceptually simpler test give false positives or false negatives in cases where the TSC test gives the correct answer?*

**Response**: This is an excellent question that touches upon some aspects of the old and new test methods that we did not elaborate on in the discussion paper. Essentially, Dr. Sacks asked whether the old PerGro test would become useful again if the model time step was set to 1 s instead of 1800 s. Our answer is "yes, but that revised test could still give false negatives in some circumstances where the TSC method gives the correct answer".

The original PerGro test is no longer useful for the default CAM5 model because even in a trusted computing environment, initial perturbations of $\mathcal{O}(10^{-14})$ K grow so rapidly that the resulting solution differences are often undistinguishable from solution differences caused by unintended code changes or incorrect porting. In our response to referee #2's comments, three reasons are listed as reasons for the rapid growth: (a) long time step, (b) state-dependent randomness in the radiation code, and (c) poorly conditioned code pieces. Reducing model time step addresses issue (a) (see Fig. 1b in our response to referee #2), thus helps to alleviate the perturbation growth; but problems (b) and (c) still exist, and lead to divergence of trusted solutions that can masks subtle but systematic solution changes. Below is an example.

We conducted PerGro test runs using 1 s time step and with radiation called every other time step (so that the time step ratio between radiation and the other parameterizations stay the same as in the default model). We then conducted simulations with the dust emission parameter changed from 0.55 to 0.45 as in the DUST case presented in the discussion paper, also with 1 s time step and with radiation called every other time step. The exercise was repeated using 11 additional sets of initial conditions. As can be seen in the figure below, the temperature RMS differences induced by the parameter change (solid orange lines) stayed substantially below the reference curves (dashed black lines) in the first ∼10 time steps, then quickly approached the reference curves

but did not exceed them in any of the ensemble members. We extended the simulations to 300 steps and the results remained the same. Based on the description of the PerGro test at http://www.cesm.ucar.edu/models/cesm1.0/cam/docs/port/pergro-test.html, one would consider the DUST case as a clear "pass", while both our TSC method and the CAM-ECT assigned the case a "fail".

It is worth noting that the PerGro method perturbs and monitors only the temperature field. Since the impact of dust emission is limited to a rather small number of grid points in very short simulations, it is not surprising that the emission change cannot be detected by PerGro even with 1 s time step. The TSC method makes use of the fact that a change in model time step directly affects all prognostic equations. We monitor multiple state variables, and also calculate the test diagnostics on land and ocean separately, thus achieved higher sensitivity with the TSC method.

**Comment:** *I'd also like clarification on the following point: On a continuum from non-answer-changing to answer-changing, I see mention of the following types of changes: (1) bit-for-bit identical, (2) answer-changing only at the round-off level, (3) answer-changing only within the limits of numerical accuracy due to the discrete time step size, and (4) climate changing, according to criteria like SIEVE or CAM-ECT. The TSC test distinguishes changes at level 3 or lower from those at level 4. But is there actually a level in between (3) and (4): changes that affect the model evolution in an appreciable way, but are not large enough to cause statistically detectable changes in climate? It seems that many bugs might fall into this intermediate regime – e.g., accidentally flipping the sign on a minor term in an equation. Do the authors feel that there is a set of changes that falls between (3) and (4), and if so, how do they expect these changes to be categorized by the TSC test?*

**Response**: This additional level between (3) and (4) might exist in principle, in which case the TSC test would assign a "fail" to the results and would not be able to distinguish them from level-(4) differences.

We also would like to point out that level-(3) and level-(4) changes are not strictly defined in a quantitative sense. For example, two simulations representing indistinguishable climate according to SIEVE based on the AMWG diagnostics package might be distinguishable using additional metrics or using CAM-ECT. Similarly, two simulations determined to be consistent using CAM-ECT based on the global and annual averages might turn out distinguishable using grid-point-wise model output and monthly time series. As for level (3), the relatively strong time step sensitivity in CAM5 implies that the numerical accuracies are substantially different when time step is changed, so level (3) is not a fixed criterion either. As can be seen in Fig. 2 of the discussion paper, if we had chosen to conduct a TSC test using a 1800 s time step instead of 2 s, the results from the RH-MIN-HIGH case (which were determined by CAM-ECT as climate-changing) would have been assigned a "pass" by TSC. In the revised manuscript, we point out these ambiguities, and clarify that while answering the "climate-changing or non-climate-changing" question using a specific set of metrics provides *one* assessment of the solution similarity/difference, the TSC method provides a different assessment of the magnitude of solution changes. From a theoretical point of view, the relationship between those two kinds of tests is not entirely clear; practically, because there are flexibilities in the design of the TSC test (e.g., time step size and pass/fail criterion), it should be possible to set up the test so that the outcome closely matches the results from a predefined climate reproducibility test. Evidences are provided in the current manuscript, and future work is planned to further evaluate the strengths and limitations of the TSC method.

[Figure]

[Figure]

Figure 1: Comparison between the temperature RMS differences caused by initial perturbation of $\mathcal{O}(10^{-14})$ K (dashed black lines, "PerGro") and the differences induced by changing the dust emission parameter from 0.55 to 0.45 (solid colored lines). All physics parameterizations used 1 s time step except for radiation which was calculated every other step. The simulations were conducted on Titan at the Oak Ridge Leadership Computing Facility using the default compiler setups. The 12 ensemble members used initial conditions sampled from different months of a previously conducted multi-year climate simulation with the default CAM5.3 model and the FC5 component set.

**Fig. 1.** Temperature RMS differences.

---

## Author Comment (AC3) · 22 Oct 2016

We thank the referee for the careful review. Our responses are detailed below.

**Comment:** *General comments:*
*Overall the paper was well-written and clear. The TSC test idea is a clever application of the time step convergence work from Wan et al. (JAMES, 2015) and appears useful. Certainly this approach is promising and inexpensive, and the manuscript is a good start. More details on the manuscript are provided below, but my main concerns to address are as follows:*
*(1) The paper would have been stronger if the test parameters had been fleshed out more thoroughly, particularly the ensemble size, the false positive rate, and number of*

*variables. For example, because this test returns a "fail" if a single variable fails, then a larger subset of variables will increase the possibility of failure by chance, so making the reader aware of this relationship would be useful.*

**Response**: The intention of this manuscript is to describe a first implementation of the TSC test procedure in the CAM5 model and to demonstrate that it is a practical and useful method for model testing. We acknowledge in the revised manuscript that the test setup can be hardened, and future work is planned to further evaluate the specific choices (e.g., ensemble size and the pass/fail criterion), and to evaluate the strengths and limitations of TSC by comparing it with other methods.

We agree that if the pass/fail criterion stays the same as described in the discussion paper, monitoring a larger set of variables will increase the possibility of failure by chance. We point out in the revised manuscript that $\mathcal{P}_0$ (i.e., the threshold $\mathcal{P}_{\min}$ for failing the test) was empirically chosen by comparing the behavior of the simulations that were expected to pass the test with those expected to fail. If the number of variables is changed, one should re-do the simulations in several trusted computing environments, calculate and plot the time series of $\mathcal{P}_{\min}$, determine the typical values, then adjust the threshold accordingly. There might be other pass/fail criteria that are less sensitive to the choice of variables, and this could be a topic for future work.

**Comment:** *(2) More details on the scope of the test would be helpful. There are bits in section 2.1 and later in 4, but it would help to better quantify the scope beyond the "equation-solving" part. In particular, the selection of variables would seem to impact the scope. Because of the limited (?) scope, an example of a bug/issue that is not caught would be helpful. And ideally this counter-example would be discussed within a larger discussion of scope as relating to the choice of variables.*

**Response**: We clarify in the revised manuscript that TSC was designed from the point of view that CAM is a general circulation model that solves a large set of differential, integral, and algebraic equations. The model variables (i.e., arrays in the code) can be

categorized into the following types:

I. Prognostic and diagnostic variables whose equations are coupled to one another, so that any change in variable $A$ will, within one time step or after multiple time steps, affect variable $B$ in the same category. Examples in this category include basic model state variables like temperature, winds, and humidity, as well as quantities calculated as intermediate products in a parameterization, for instance the aerosol water content (which affects radiation and eventually temperature), and the convective available potential energy (which affects the strength of convection hence temperature and humidity).

II. Prognostic variables that are influenced by type-I variables but do not feedback to type I. An example could be passive tracers carried by the model to investigate atmospheric transport characteristics (e.g., Kristiansen et al., 2016).

III. Diagnostic quantities that are calculated to facilitate the evaluation of a simulation, but do not feedback to type I. Examples include the daily maximum 2-m temperature, the ice-to-liquid conversion rate in the cloud microphysics parameterization (which is a quantity calculated merely for output in CAM5.3), and any variable specific to the COSP simulator package (Bodas-Salcedo et al., 2011).

Code pieces in the model can be categorized accordingly.

Our standpoint is that the essential characteristics of the simulated climate are determined and represented by type-I variables, and the TSC test is designed for code pieces in this category. Since all variables in this type are coupled, and since our test method monitors instantaneous and grid-point-wise values before chaos sets in, any significant bug or compiler error (that affects the solution of the *coupled* equation set) should be detectable through the monitoring of a single variable, as long as there is sufficient integration time for the impact to evolve to a discernable magnitude and propagate to that variable. When the simulations are short (for instance on the order

of minutes of model time as in TSC), tracking multiple variables can help increase the sensitivity of the test (decrease the chance of false negative) since discernable solution differences might show up earlier in some variables than in others.

The list of variables monitored by TSC can be extended to type-II variables defined above, if the user wishes to cover the related code pieces in the testing. Diagnostic variables of type I or type II should not be included in the list because the concept of time step convergence does not apply. Consequently, bugs in the implementation of any "diagnostic-only" calculations, e.g., a satellite simulator, would not be detected by TSC. Also, issues with code pieces that are not exercised, for instance the restart capability, would not be caught by the test either.

**Comment:** *(3) The experimental results section would be stronger if the experiments more closely represented the stated scope (see previous comment). Then the reader would gain a better understanding of the tool's utility. The chosen experiments are essentially the same as those in Baker, et al. (GMD, 2015). While it is important to include those, it is not clear that the 10 variables chosen would be sufficient to catch errors in all parts of the code (as stated in section 2.1), so it would be helpful to have an example of an error that is not caught. Also, several times (e.g., section 2.1) "code modifications" are mentioned as an application for this test, but there is not an example supporting this statement (and including such an example seems important).*

**Response**: Please see our response to the previous comment for a clarification on the scope of our test, and for examples of bugs/issues that would not be caught by TSC.

As for the "code modifications", two examples from Milroy et al. (2016) that represent code optimization strategies are included in the revised manuscript: "division-to-multiplication" (DM) and "precision" (P).

**Comment:** *(4) Regarding the TSC's use of the t-test, please clarify the reason for the directional t-test. In particular, why does the test only check if the mean is larger than zero? (i.e., $\mu_j > 0$) - as opposed to the non-directed alternative hypothesis:*

*mu_j / = 0. Certainly mu_j can be negative, so is this scenario just not of concern? For example, in figure 3, if delta_RMSD for variable was negative for all 12 members, then TSC would issue a pass. I am not necessarily questioning the efficacy of the test procedure, but I have to wonder if systematically negative results can be problematic as well or even indicative of an issue with the simulation being tested.*

**Response**: The test metric of the TSC method is the model's time stepping error in simulations conducted with 2 s time step compared to trusted reference solutions conducted with 1 s time step. If both the model equations and the discretization methods stay the same, the time stepping error is expected to stay the same. If bugs are introduced, or if the code is not compiled or executed correctly, the resulting numerical integration will not be solving the originally intended equations, thus not converging to the original reference solutions, resulting in larger apparent time stepping errors. In a non-answer-changing case, while $\Delta$RMSD can be negative by chance for an ensemble member, it is very unlikely that it will be negative for all members. The only situation we could imagine systematically negative $\Delta$RMSD to occur would be the implementation of a new and more accurate set of time stepping algorithms that featured smaller sensitivity to the step size change of 1 s to 2 s, but yet produced very similar solutions at 1 s time step when compared to the original code. Such a case of algorithm update would be considered a substantial code change, so methods like TSC and PerGro would not be the most natural tests to perform since they are designed to assure that the solutions are unchanged. Once the merits of new algorithms have been confirmed and a new default model is established, a new set of reference solutions (with 1 s time step) and trusted solutions (with 2 s time step) should be generated and used for future testing. This is explained in the revised manuscript.

**Comment:** *Specific comments:*
*(1) Section 1: line 20: Check the use of "reproducibility" in this context.*
*(2) Section 1: The first couple paragraphs are quite similar in parts to the text in Baker, et al. (GMD, 2015), including the same references and some of the same phrases,*

*which is a bit awkward.*

**Response**: The first two paragraphs of the manuscript have been rewritten.

**Comment:** *(3) page 3, line 24: The tool's application to "code modifications" is mentioned here and in section 5, but I don't believe this is being tested in the experiments. It may be of interest to look at CESM code modification experiments in the followup to Baker, et al. (GMD, 2015), which is:*
*Daniel J. Milroy, Allison H. Baker, Dorit M. Hammerling, John M. Dennis, Sheri A. Mickelson, and Elizabeth R. Jessup, "Towards characterizing the variability of statistically consistent Community Earth System Model simulations." Procedia Computer Science (ICCS 2016), Vol. 80, 2016, pp. 1589-1600.*
*(http://www.sciencedirect.com/science/article/pii/S1877050916309759)*

**Response**: Thanks for the reference. Two examples of code modification from Milroy et al. (2016) that represent code optimization strategies are included in the revised manuscript: "division-to-multiplication" (DM) and "precision" (P).

**Comment:** *(4) Section 2.1: I would really like to better understand how the selection of the 10 variables affects (or does not affect) the scope.*

**Response**: Please see our response to general comment #(2) for a categorization of the model variables. The TSC method described in the manuscript is designed to test all code pieces that affect type-I variables, and the 10 variables we chose all belong to that type. Monitoring more (fewer) variables of the same type would not affect the scope of the test but could affect the test's sensitivity for a chosen integration length, i.e., it could decrease (increase) the chance of false negative, since bugs or issues associated with a specific piece of code might take longer time to cause discernable solution differences in one variable than in another. Adding type-II variables, on the other hand, would extend the scope of the TSC test.

**Comment:** *(5) page 2, line 25: This is not exactly true as CAM-ECT has been used*

*to pinpoint errors in specific code modules (e.g. FMA error on Mira detailed in Milroy et al. 2016).*

**Response**: The respective sentence in the discussion paper read "Moreover, since each ensemble member is a one-year simulation, it is unlikely that the method can be used to test a small subset of the model components, or a code that is still in debugging stage thus numerically unstable for long simulations (criterion 5)." The statement is now revised. Based on the categorization of model variables discussed earlier in our response to general comment #(2), we expect CAM-ECT to be capable of pinpointing issues associated with variables of type II and type III. The FMA error on Mira as described in Milroy et al. (2016) is an interesting case worth further investigation. To keep the manuscript focused, we do not include any detailed discussions on that topic, but some of our thoughts are included here:

Based on our definition of the type-I variables, we do not expect CAM-ECT to be able to pinpoint issues in code pieces that affect the calculation of type-I variables. The argument is that since all variables in this type are inherently coupled, any substantial change in one equation should have affected all the type-I variables after a year of model integration. In the Milroy et al. (2016) paper, it was reported that six output variables from the CAM model were identified as suspects for further inspection. We contacted the authors and obtained the actual list of those variables. Five out of those were in fact type-III ("diagnostic-only") variables as we suspected, but it was curious that the sixth variable was CLDLIQ, the mass concentration of liquid-phase condensate in stratiform clouds. Given the important role of this prognostic variable in the model, it is counterintuitive to us that values of this variable obtained on Mira were inconsistent with the control ensemble while values of other closely related variables like temperature, humidity, and cloud properties were consistent. Could it be that the inconsistency in CLDLIQ was very minor thus the impacts on other variables were negligible? Would we see more substantial inconsistencies and in more variables if spatial patterns were included in CAME-ECT? The answers to these questions are unknown

at this point. We also learned from Mr. Milroy and Dr. Baker that a number of code lines and local variables in the cloud microphysics parameterization were identified as being affected by FMA. It was again counterintuitive to us that those local variables included the microphysical tendencies of cloud droplet and ice crystal number concentrations, but the corresponding state variables were deemed consistent between the Mira results and those from the trusted computers. To us, this again indicates that the case is worth further investigation in the future.

**Comment:** *(6) page 3, lines 27-28: Regarding "...when the accuracy limits related to the algorithmic implementation are taken into account." This doesn't appear to be considered in the rest of the paper.*

**Response**: The subsection on test scope has been rewritten. What we meant by the sentence cited above has been rephrased: From the point of view that CAM is a general circulation model that solves a large set of differential, integral, and algebraic equations, we consider the results as unchanged if the numerical solutions are found to have the same time stepping error when compared to a predefined set of reference solutions.

**Comment:** *(7) page 4, line 14: I agree with #5 as a desirable feature, but I don't believe that evidence was given in this manuscript that TSC fulfills #5. Certainly no evidence was given in Baker, et al. (GMD, 2015) that CAM-ECT satisfies #5, though one can imagine the framework could possibly apply. So if the claim is that TSC fulfills this while CAM-ECT does not, it would be stronger to provide specific evidence of such a case for TSC (i.e., an experiment to validate the claim).*

**Response**: In the earlier study of Wan et al. (JAMES, 2015), in addition to assessing time step convergence in the the full CAM5 model, convergence analysis was also done for configurations that exercised the dynamical core plus only one parameterization or parameterizations group at a time, e.g., deep convection, shallow convection, large-scale condensation, or the stratiform cloud microphysics. This was an attempt to

find out which of those parameterizations led to the convergence rate of 0.4 (instead of 1) in the full model. Simulations were also conducted using the dynamical core plus a very simple saturation adjustment scheme, or with the cloud microphysics parameterization of CAM5 but with the formation and sedimentation of rain and snow turned off (see Figure 3 in Wan et al., 2015, JAMES). Those simulations conducted with a small portion of the CAM5 code were likely to blow up if the integration had proceeded longer than a few hours or days, and certainly would not produce any realistic climate, but they clearly revealed different convergence rates and time step sensitivities associated with different components of the model code. We imagine the same strategy of breaking down the code into small exercisable units and evaluating convergence could be used to pinpoint bugs when, e.g., a code refactoring leads to unexpected failing results from the TSC test. This is why we believe the TSC method fulfills feature #5, and we clarify it in the revised manuscript.

**Comment:** *(8) Section 2.3: Since the starting conditions for the TSC ensemble are samples from "a previously conducted long-term simulation", does one need to update this simulation with answer-changing CESM tags, for example? Also does "long-term" mean 1-year or ????? Please give more details on how this part of the process works.*

**Response**: We clarify in the revised manuscript that the initial conditions for individual ensemble members were samples from the first year (after 6 months of spin-up) of a previously conducted 5-year simulation. In our experience, climate simulations of 1–5 years are frequently carried out during model development or evaluation, making such initial conditions easy to obtain. The basic requirements for the initial conditions for TSC are that (i) they contain reasonably spun-up values for the model state variables (e.g., not all zeros or spatially constant values for the hydrometeors or aerosol concentrations), and (ii) they represent synoptic weather patterns in different seasons. Those initial conditions do *not* need to represent well-balanced states in the quasi-equilibrium phase of a multi-year climate simulation. In fact, the default model time step of 1800 s was used when creating the initial conditions for this study, while the control and test

simulations in TSC used 1 s or 2 s time step, so the model state was certainly not well-balanced during those TSC simulations. We think the same set of initial conditions can be used after answer-changing code tags are established – until a point when the list of prognostic variables in the model becomes substantially different. Then it would be useful to regenerate the initial conditions, and rethink which variables should be monitored by the test.

**Comment:** *(9) Would the TSC test results be affected if the ensemble was created instead by perturbing initial conditions (since this does not require a previous simulation)?*

**Response**: We have not tried this idea yet, but expect that the answer would depend on the magnitude of the initial perturbations. Since our intended simulation length is on the order of minutes to an hour, small perturbations like those used in PerGro and CAM-ECT would not have time to trigger sufficient spread (variability) among the ensemble members. The need for ensemble was demonstrated by Figure 3 in the discussion paper.

**Comment:** *(10) page 5, last paragraph: Should point out that this test (RH-MIN-HIGH from .8 to .9) is from Baker, et al. (GMD, 2015) for comparison.*

**Response**: Done.

**Comment:** *(11) page 5, line 34-page 6, line 1: "[...] concept of self-convergence since no structural changes [...] have been introduced into the model." More generally (and relevant to the discussion in Sect 3.2), what if the modified model's 2s timestep behavior is closer to the 1s timestep reference model than to itself for 1s timestep? In other words, what if its convergence behavior to the reference model is different than its self-convergence?*

**Response**: Given the complexity of the model and its time stepping algorithms, we would argue it is very unlikely that a modified model's behavior at 2 s time step will be

closer to the reference solution at 1 s of an old model than to the reference solution at 1 s time step of the new model. As mentioned earlier in our response to general comment #(4), the only situation we could imagine to see that kind of results would be the implementation of a new and more accurate set of time stepping algorithms that featured smaller sensitivity to the step size change of 1 s to 2 s, but yet produced very similar solutions at 1 s time step when compared to the original code. Such a case of algorithm update would be considered a substantial code change, so methods like TSC and PerGro would not be the most natural tests to perform since they are designed to assure that the solutions are unchanged.

**Comment:** *(12) page 6, line 13: The "more substantially" comment is a bit vague. The change is already labeled "climate-changing", which itself seems substantial. Certainly this change is more substantial than, for example, changing the order of operations in the code or something similarly "minor". Clarify?*

**Response**: Two simulations that are both "climate-changing" can differ from the control simulation by different magnitudes. We revised the wording of the respective sentences as follows:

"If we had introduced larger changes in the model, e.g., by changing cldfrc_rhminh to 0.999 instead of 0.9 from the default value of 0.8, or by replacing a certain parameterization by a different scheme, the impact might be more visible at the default step size. In contrast, if the parameter change were smaller, e.g., from 0.8 to 0.82 instead of 0.9, the red and blue convergence pathways in Fig. 2 might not diverge until a step size on the order of a few seconds."

**Comment:** *(13) page 7, first paragraph: Did the authors use the SIEVE method to verify all of the results presented? It is not clear. Also wondering if the example (NU) in Baker, et al. (GMD, 2015) that passed (but that Baker et al. claim should have failed) was independently verified by the authors with SIEVE?*

**Response**: We point out in the revised manuscript that SIEVE is not necessarily a

satisfactory procedure or gold standard due to the ambiguity in the criteria for "climate-changing". For example, two simulations judged to be indistinguishable by SIEVE based on the AMWG diagnostics package might be distinguishable using additional metrics or using CAM-ECT. In the revised manuscript, we clarify that we did not independently verified any of the parameter perturbation examples from the Baker et al. (GMD, 2015) paper. Rather, based on the magnitudes of the parameter changes and our understanding of the mechanisms through which those parameters affect the simulated atmospheric motions, we expected all those changes to cause solution differences discernable by TSC.

**Comment:** *(14) Followup to (12): Recommend that the authors come up with another example of a small scale change that CAM-ECT would not catch because of its use of the global and annual mean (but that TSC would) - other than the NU test from Baker et al. (GMD, 2015). This would probably have to be more subtle than the experiments in Baker, et al. (GMD, 2015). I think this recommendation is particularly pertinent given the list of desired features on page 2 (and that TSC should achieve #6 while CAM-ECT will not).*

**Response**: Since the test diagnostics of TSC are calculated from instantaneous grid-point-wise model output while CAM-ECT uses global and annual averages, we believe it is reasonable to expect that the former has a larger chance to catch regional differences in the solutions. The NU case has provided evidence to support this reasoning. It is worth noting we also stated in the discussion paper that

"On the other hand, since a large number (120) of model output variables are used in CAM-ECT and the simulations are relatively long (1 year), the chance of missing a climate-changing modification (i.e. getting a false 'pass') is relatively small."

We agree that further examples of small-scale solution changes would be informative, but they would not affect the key messages we are trying to deliver in this manuscript. In a more generally sense, it would be useful to compare TSC with CAM-ECT using

additional (more subtle and challenging) test cases so as to further understand the strengths and limitations of either method. We would be delighted to collaborate with the CAM-ECT developers on that.

**Comment:** *(15) page 7, line 16: A false negative example would be a great addition and improve the reader's understanding of the tool's scope.*

**Response**: As mentioned earlier in the response to general comment #(2), any bug in "diagnostic-only" parts of the model code, e.g., the calculation of daily maximum 2-m temperature, or the implementation of a satellite simulator, would not be caught by TSC. Another type of false negative is discussed in our response to the next comment.

**Comment:** *(16) Section 3.2: The splitting into the two domains could be explained more (it is discussed a bit again later in 4.1). It seems a bit arbitrary and suggests that the DUST and CONV-LND failures cannot be detected otherwise. One issue is that by splitting into domains (effectively doubling the number of variables), the false positive rate is being increased. Would be helpful to have more guidance on variable selection and limitations.*

**Response**: Without splitting the domain into land and ocean we would indeed get false negative results in the both cases (DUST and CONV-LND). If we had not chosen to include the aerosol number concentrations in the list of monitored variables, the DUST case would also end up being a false negative. In these cases, the false negative results might be avoidable if the simulations were considerably longer. Limited sensitivity is a price one might have to pay in exchange for the rather low computational cost. We think the test setup described in the manuscript is a practically useful choice if the user is primarily interested in the basic atmospheric state and clouds and aerosols. If the model was going through an active development in, say, the representation of atmospheric chemistry, it would be beneficial to add concentrations of some chemical species to the list of monitored variables. The TSC method is flexible in this regard, although we would like to emphasize again that only prognostic variables of type I and

type II defined in the response to general comment #2 can be used for the test diagnostics. The concept of time step convergence does not apply to variables that are not calculated using an evolution equation.

As for the impact of the number of variables on the false positive rate, we clarify that the $\mathcal{P}_0$ in this manuscript (i.e., the threshold $\mathcal{P}_{min}$ for failing the test) was empirically chosen by comparing the behavior of the simulations that were expected to pass the test with those expected to fail. If the number of variables is changed, one should redo the simulations in several trusted computing environments, calculate and plot the time series of $\mathcal{P}_{min}$, determine the typical values, then adjust the threshold accordingly. There might be other pass/fail criteria that are less sensitive to the choice of variables, and this could be a topic for future work.

**Comment:** *(17) page 8, lines 16-28: I'm still struggling a bit with understanding the scope, which is discussed again here in terms of what will and won't be caught (e.g., aerosol concentrations). Please clarify earlier and consider including supporting experimental results.*

**Response**: Revision is made in line with our responses to general comment #(2) and specific comments #(15)-(16).

**Comment:** *(18) page 9, line 10: If the implicit assumption that the random variables (mu-sub-j) are Gaussian distributed is violated, will the TSC test results be affected? (And has this been explored? An example could be something like truncation...)*

**Response**: We have not explored this. The manuscript only describes the first implementation of TSC and provides evidences that this is a useful method. The test setup can be hardened in the future.

**Comment:** *(19) page 9, Step 3 (line 30 -> page 6): More clarification is needed here. For the t-test, the choice of .05% is conservative (as acknowledged in text), and it is clear that the specified t-statistic (4.437) is dependent on both the .05% cutoff \*and\**

*the sample (ensemble) size (M=12). However, there is a less intuitive dependence on the number of variables that should be pointed out (and discussed). Because the t-test is performed on each variable \*individually\*, then the number of variables examined certainly affects the overall test failure rates. The conservative choice of .05% may make sense for the 20 variable subset (meaning that a single variable has to fail quite badly to cause a failure of the overall test). However, if one were to use 2 variables (or 100 variables), the .05% may no longer be the best choice. I think this should be addressed given the discussion on page 8 (line 25) that one could choose to include more (and presumably fewer) fields.*

**Response**: We agree with the referee's comment. As mentioned above, we clarify that the $\mathcal{P}_0$ in this manuscript (i.e., the threshold $\mathcal{P}_{\min}$ for failing the test) was empirically chosen by comparing the behavior of the simulations that were expected to pass the test with those expected to fail. If the number of variables is changed, one should re-do the simulations in several trusted computing environments, calculate and plot the time series of $\mathcal{P}_{\min}$, determine the typical values, then adjust the threshold accordingly. There might be other pass/fail criteria that are less sensitive to the choice of variables, and this could be a topic for future work.

**Comment:** *(20) page 9, line 9): Please clarify the reason for the directional t-test and consider updating/clarifying the accompanying discussion on page 9,line 25 -> page 10, line 4.*

**Response**: Done. See also our response to general comment # (4) above.

**Comment:** *(21) page 9, line 10: A minor point, but technically one cannot "accept" the null hypothesis. (One can fail to reject the null hypothesis or reject it.)*

**Response**: Corrected. We now use "fail to reject".

**Comment:** *(22) page 11, line 1: Was the .89 vs. .897 detectable by SIEVE? Also how long of a simulation was run for SIEVE in this case?*

**Response**: 10-year simulations were conducted in both cases and compared to a control simulation using the AMWG diagnostics. The case of 0.897 was indistinguishable from the control by SIEVE using the standard plots, but given the rather direct impact of this parameter on the cloud formation in the model, we thought the difference might be detectable by additional metrics. The expected "fail" was rather an educated guess that was later confirmed by TSC. This is clarified in the revised manuscript.

**Comment:** *(23) page 10: Given the FMA issues found for Mira in Milroy et al. 2016 (and also for BlueWaters), I am questioning the Cori results a bit - also because the results in Table 1 for Cori are not as definitive as for the other machines. Cori uses FMA by default, or was it disabled for these experiments? How long were the simulations examined by SIEVE for Cori?*

**Response**: As mentioned earlier in the response to specific comment #(5), we think the FMA issue is an interesting one worth further investigation. There is the possibility that the impact of FMA is far below the magnitude of the time stepping error in very short simulations thus not detectable by the TSC setup described in the manuscript. In that case, the use of multiple test methods might help better understand the impact of the FMA issue from different angles. Since the case is not yet well understood, and the Cori example is not essential for demonstrating the basic idea and utility of the TSC method, we do not show the Cori results in the revised manuscript.

A separate comment on the Cori result: in Table 1 the $\mathcal{P}_{\min}$ values were shown only at 5 minutes and 30 minutes after model initialization. While the two numbers from Cori were indeed less definitive than those from the other machines, from the complete time series shown in Figure 6 of the discussion paper, the Cori results seem less suspicious. This made us realize that "pass/fail" criteria based on results at a single time instance are more likely to lead to false positives and negatives. Since the model equations are evolution equations, and the TSC method looks at a time window well within the deterministic forecast limit, it might be beneficial to determine pass or fail using model results from more time steps within a certin time window.

**Comment:** *(24) page 13, line 25: " ...failing the test will very likely mean the climate will be different. Passing the convergence test should hence be considered a necessary condition..." I don't quite agree with this. Many of the parameterizations could be quite different in the short term (because of sensitivity), but the longer term behavior is basically the same. In other words, the weather after 150s may look different (e.g., raining or not), but the annual climate is the same. (This assertion is also made on page 7, lines 15-16)*

**Response**: It sounds like the referee was thinking about chaos and predictability. Since the TSC test only looks at a time window of a few minutes to an hour, We believe the problem should be sufficiently deterministic.

**Comment:** *Technical Corrections*
*(1) page 2, line 3: Remove the final word "did" from the sentence.*
*(2) page 2, line 29: The second occurence of "simulation" should be plural.*
*(3) page 3, line 9: Spell out the number 3 (three).*
*(4) page 3, lines 23-25: Consider breaking this sentence into smaller parts.*
*(5) page 5, line 8: "thus saves" should be "thus saving".*
*(6) page 5, line 18: "dependences" should be "dependencies".*

**Response**: All corrected or revised.

**Comment:** *Final thoughts. I like the idea of this work, and I hope that the comments and suggestions provided will be helpful for the revision of the paper. I believe that more flushed out algorithm details, a clarification of scope, and better alignment of the experimental results with the stated features of the test will strengthen the paper and its impact and utility.*

We thank the referee for the detailed and very helpful review. The questions and suggestions, together with the comments from the other referee and from Dr. Sacks, prompted us to think deeper about our method. We have made substantial revisions in the manuscript to clarify the purpose and scope of our method as well as our understanding of its relationship to other methods. We also added discussions on the details of the test design, including the ensemble size, test diagnostics, and pass/fail criterion, and acknowledge that these can be further evaluated and hardened. We intend to continue this work and obtain more comprehensive understanding of the strengths and limitations of the TSC method.

---

## Author Response (AR1)

Dear Editor,

We hereby submit a revised version of the manuscript gmd-2016-142 entitled "A new and inexpensive non-bit-for-bit solution reproducibility test based on time step convergence (TSC1.0)" for consideration of publication in GMD. We appreciate the careful and insightful reviews from the anonymous referees and from Dr. W. Sacks. In response to the comments and suggestions, we have made the following changes in the manuscript:

- 1. The purpose of the proposed testing method is clarified. We have realized that it is more accurate to state that the TSC test is designed for regression testing, i.e., for verifying that results from a model stay the same despite changes in the code or in the computing environment (Sect. 1). The TSC method considers the outcome of a simulation unchanged if the numerical solution is found to have the same time stepping error relative to a reference solution obtained with a previously verified code and computing environment (Sect. 2.1). Our understanding of the linkages and distinctions between TSC and other testing methods is explained in Sect. 5.3.
- 2. The scope of the proposed method is clarified. We point out in the Abstract and explain in Sect. 2.1 that the TSC method is designed for identifying numerically significant changes in solutions of evolution equations. It does not detect issues associated with diagnostic calculations that do not feedback to the model state variables.
- 3. More information is provided in Sect. 2 and Sect. 3 on the reasoning behind the specific choices we made for the version 1.0 implementation, for example the list of monitored variables, the splitting of model domain into land and ocean, the pass/fail criterion, and the initialization strategy. We also clarify that many of those choices are practical and empirical, and can be further evaluated and improved in the future (Sect. 3.2 and Sect. 5.3).
- 4. The overall pass/fail criterion is revised (Sect. 3.2). The use of multiple time steps instead of a single time instance reflects our perspective of viewing the model integration as a time evolution problem. We also point out in the manuscript that the revised pass/fail criterion is still empirical and preliminary, and can be further evaluated in the future (Sect. 3.2 and Sect. 5.1).
- 5. Two test cases with code modifications following Milroy et al. (2016) are added. Three cases with perturbed parameters and two cases with change of computing environment are removed. The purpose is to focus the discussion of the result on comparison with CAM-ECT.
- 6. All simulations presented in the discussion paper have been repeated with the radiation parameterization calculated every other time step instead of only at the beginning of the simulations. We found this change to have only very small impact

on the outcome of the TSC test. Nevertheless, when evaluating the TSC methodology using different test cases (Sect. 4), we present the new results so that the time step ratios between different model components remain the same as in the default model despite the change in time step sizes. In Sect. 2.3 where the concept of time step convergence is introduced, we present the old results for consistency with the earlier work of Wan et al. (2015), but add a note that the calling frequency of radiation does not change the convergence property of the full CAM5 model.

- 7. A brief discussion (Sect. 5.2) is added on the impact of noisy parameterization on the results of the TSC test.
- 8. Reasons for the rapid growth of initial perturbation in the CAM5 model are summarized in Sect. 1.
- 9. Typographical and grammatical errors are corrected at miscellaneous places.

Our detailed responses to the reviewers' comments and the corresponding changes in the manuscript are attached in the next pages.

Sincerely, Hui Wan

**Reply to Dr. W. Sacks**

We thank Dr. Sacks for his insightful questions. Our responses are detailed below.

**Comment:** This is a clever idea, and the paper is very well written. I'd like to be convinced that this technique truly has more power than seemingly simpler techniques. For example, can some of the same experiments be redone with this set of runs?:

(1) control: unmodified model with 1s time step

(2) baseline for comparison: unmodified model with 1s time step, with a roundoff-level perturbation in the temperature field

(3) test code: some change in the code with 1s time step

Basically, I'd like to be convinced that the "time step convergence" is truly needed here, and that it truly provides more power than just comparing two versions of the model with a short time step. Does the above, conceptually simpler test give false positives or false negatives in cases where the TSC test gives the correct answer?

**Response**: This is an excellent question that touches upon some aspects of the old and new test methods that we did not elaborate on in the discussion paper. Essentially, Dr. Sacks asked whether the old PerGro test would become useful again if the model time step was set to 1 s instead of 1800 s. Our answer is "yes, but that revised test could still give false negatives in some circumstances where the TSC method gives the correct answer".

The original PerGro test is no longer useful for the default CAM5 model because even in a trusted computing environment, initial perturbations of  $\mathcal{O}(10^{-14})$  K grow so rapidly that the resulting solution differences are often undistinguishable from solution differences caused by unintended code changes or incorrect porting. In our response to referee #2's comments and in the introduction section of the revised manuscript, three reasons are listed as reasons for the rapid growth: (a) long time step, (b) state-dependent randomness in the radiation code, and (c) particular code pieces. Reducing model time step addresses issue (a) (see Fig. 2b below in our response to referee #2 and Fig. 1b in the revised manuscript), thus helps to alleviate the perturbation growth; but problems (b) and (c) still exist, and lead to divergence of trusted solutions that can masks subtle but systematic solution changes. Below is an example.

We conducted PerGro test runs using 1 s time step and with radiation called every other time step (so that the time step ratio between radiation and the other parameterizations stay the same as in the default model). We then conducted simulations with the dust emission parameter changed from 0.55 to 0.45 as in the DUST case presented in the discussion paper, also with 1 s time step and with radiation called every other time step. The exercise was repeated using 11 additional sets of initial conditions. As can be seen in Fig. 1 below, the temperature RMS differences induced by the parameter change (solid orange lines) stayed substantially below the reference curves (dashed black lines) in the first ~10 time steps, then quickly approached the reference curves but did not

---

## Author Response (AR2)

**Reply to Referee #1**

**H. Wan**
**January 16, 2017**

Our responses are listed below:

*General comments:*

*Overall this revision is much stronger than the original and the presentation is improved. The addition of information on the PERGRO test and its limitations in the context of CAM5 will be useful for many readers. In addition, I think the revised pass/fail criteria makes more sense in the context of time integration, and look forward to further analysis in the future.*

*My first three main concerns (numbered as before) have been adequately addressed:*

*(1) The method itself is now better described, and I appreciate the addition of Section 5. Several issues are now noted as being planned future work.*

*(2) The authors have done a good job clarifying the scope and applicability of the method (e.g., section 2.1). I have a better understanding of its utility.*

*(3) The experimental results section now better aligns with the scope (some experiments have been added and some deleted - and the discussion improved).*

We appreciate the positive feedback, and thank the referee for the assessment.

*The response to concern (4) regarding the one-sided t-test still does not entirely make sense to me." While I agree with this sentence in the response (page 9): "In a non-answer-changing case, while $\Delta RMSD$ can be negative by chance for an ensemble member, it is very unlikely that it will be negative for all members", the statement may also be true if you replaced the word "negative" with "positive". So that leads to the question of what makes a positive deviation more likely than a negative deviation?*

*I \*think\* the assumption being made is that a bug in the code could not somehow make the time stepping more accurate - which is what is needed to make $\Delta RMSD$ negative. While that makes sense from a theoretical standpoint, in practice (because this is a chaotic floating point simulation), wouldn't it be safer to use the non-directed t-test? Further, the explanation in the response (bottom of page 9) that negative $\Delta RMSD$ should not be tested with this method because it's solution-changing, strengthens my opinion that a fail should be issued in this case:*

*"The only situation we could imagine systematically negative $\Delta RMSD$ to occur would be the implementation of a new and more accurate set of time stepping algorithms that featured smaller sensitivity to the step size change of 1 s to 2 s, but yet produced very similar solutions at 1 s time step when compared to the original code. Such a case of algorithm update would be considered a substantial code change, so methods like TSC and PerGro would not be the most natural tests to perform since they are designed to assure that the solutions are unchanged."*

The following sentences are added to Section 3.2 of the manuscript:

Let us use the symbol $S_{\mathrm{ori,1s}}$ to denote the reference solution of the original equation set obtained with a 1 s time step, and use $S_{\mathrm{test,2s}}$ to denote a test simulation conducted with the new equation set using a 2 s time step. The RMSD calculated in TSC is the root-mean-square of $(S_{\mathrm{test,2s}} - S_{\mathrm{ori,1s}})$ which can also be expressed as

$$(S_{\mathrm{test,2s}} - S_{\mathrm{test,1s}}) + (S_{\mathrm{test,1s}} - S_{\mathrm{ori,1s}}) \tag{5}$$

The difference in the first pair of parentheses in (5) measures the time-step sensitivity of the solution of the new equation set, while the difference in the second pair of parentheses measures the discrepancy between the reference solutions of the old and new equation sets. By using a one-sided test, we assume that the second difference will be non-negligible, and that the two differences will not incidentally compensate each other to result in values of $(S_{\mathrm{test,2s}} - S_{\mathrm{ori,1s}})$ that are systematically smaller than $(S_{\mathrm{ori,2s}} - S_{\mathrm{ori,1s}})$. The validity of this assumption can be evaluated in the future by comparing TSC results using one-sided and two-sided tests.
* * *
*Specific comments:*

*-Is TSC 1.0 available to download or is it only available by request from the authors?*

We state in the "Code and data availability" section that the scripts are available from the corresponding author.
* * *
*Technical Corrections*

*p.3, line 28: "characteristics" => "characteristic"*
*p.11, line 23: Missing article at the beginning of the sentence "One-sided test is ..."*

Corrected.

[revised manuscript text omitted]